# Causal Structure Recovery with Latent Variables under Milder Distributional and Graphical Assumptions

**Xiu-Chuan Li**[1]     **Kun Zhang**[2,3]     **Tongliang Liu**[1]
[1]Sydney AI Centre, University of Sydney
[2]Carnegie Mellon University
[3]Mohamed bin Zayed University of Artificial Intelligence

## Abstract

Traditional causal discovery approaches typically assume the absence of latent variables, a simplification that often does not align with real-world situations. Recently, there has been a surge of causal discovery methods that explicitly consider latent variables. While some works aim to reveal causal relations between observed variables in the presence of latent variables, others seek to identify latent variables and recover the causal structure over them. The latter typically entail strong distributional and graphical assumptions, such as the non-Gaussianity, purity, and two-pure-children assumption. In this paper, we endeavor to recover the whole causal structure involving both latent and observed variables under milder assumptions. We formulate two cases, one allows entirely arbitrary distribution and requires only one pure child per latent variable, and the other requires no pure child and imposes the non-Gaussianity requirement on only a subset of variables, and they both avoid the purity assumption. We prove the identifiability of linear latent variable models in both cases, and our constructive proof leads to theoretically sound and computationally efficient algorithms.

## 1 Introduction

Understanding causal relations is a fundamental element of artificial intelligence (Schölkopf et al., 2021; Schott et al., 2018; Dominguez-Olmedo et al., 2022; Wang et al., 2021; Yao et al., 2021; Lin et al., 2023; Huang et al., 2023; Hong et al., 2024). The gold standard is to use randomized experiments, but this is usually too expensive or even impractical. Therefore, there has been significant attention towards revealing causal relations through analysis of observational data, commonly known as causal discovery. Most traditional approaches focus on the situation without latent variables, such as constraint-based PC algorithm (Spirtes et al., 2000), score-based Greedy Equivalence Search (GES) (Chickering, 2002), and some Functional Causal Model-(FCM-)based algorithms (Shimizu et al., 2006; 2011; Hoyer et al., 2008a; Zhang & Hyvarinen, 2009; Peters et al., 2014; Mooij et al., 2016). However, in complex systems, we typically fail to collect and measure all task-relevant variables. Many algorithms have been proposed to handle the situation with latent variables, such as constraint-based Fast Causal Inference (FCI) (Spirtes et al., 1995), score-based Greedy PAG Search (GPS) (Claassen & Bucur, 2022), and also some FCM-based algorithms (Hoyer et al., 2008b; Tashiro et al., 2014; Salehkaleybar et al., 2020; Maeda & Shimizu, 2020; Cai et al., 2023).

While the above approaches can reveal causal relations between observed variables with or without latent variables, they cannot identify latent variables, let alone infer their causal relations. However, researchers may care more about the causal structure over latent variables in many cases (Silva et al., 2006). Assuming linear causal relations and no observed variable being the parent of any latent one in the underlying causal graph, many existing works employed sparsity of causal edges to facilitate latent causal structure learning. For instance, some early works (Silva et al., 2006; Kummerfeld & Ramsey, 2016) have proven that latent causal structure can be recovered under the *three-pure-children assumption* that each latent variable has at least three *pure children* (an observed variable $O$ is called a pure child of a latent variable $L$ if $O$ has no child and only one parent $L$, see Definition 2.). Others (Cai et al., 2019; Xie et al., 2020; 2022) have relaxed the three-pure-children assumption to

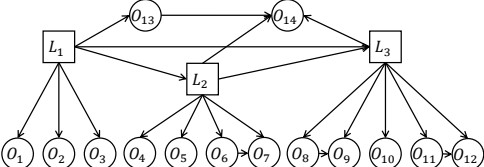 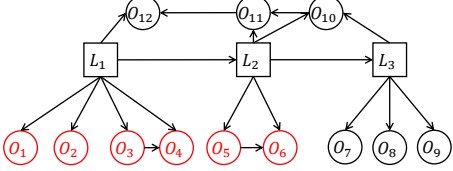

(a) $\mathcal{G}_1$: $L_3$ has only one pure child and each noise has arbitrary distribution.

(b) $\mathcal{G}_2$: $L_2$ has no pure child and only noises of red variables are non-Gaussian.

Figure 1: The whole causal structure can be recovered in both the above cases where none of the purity, non-Gaussianity, and two-pure-children assumption holds.

the *two-pure-children assumption* that each latent variable has at least two pure children. However, they entailed two additional assumptions: the *purity assumption* that there is no causal edge between observed variables and the *non-Gaussianity assumption* that noises of all variables are non-Gaussian. On this basis, Xie et al. (2023b) made a further step by eliminating the purity assumption.

In the real world, since some variables might have nearly Gaussian distributions (Lyon, 2014), the non-Gaussianity assumption might not hold. Besides, some observed variables may directly influence others, violating the purity assumption, e.g., in financial markets, while stock returns may be confounded by some economic or political factors, they may also be causally related (Adams et al., 2021). Moreover, the occurrence of pure children will become less frequent without the purity assumption. Finally, when observed variables are also causally related, we usually want the whole causal structure involving both latent and observed variables rather than only the latent causal structure. Therefore, we endeavor to recover the whole causal structure in the case where none of the non-Gaussianity, purity, and two-pure-children assumption holds.

Recovery of the whole causal structure requires us to first identify latent variables and then infer causal relations between any two variables. Existing works typically identify a latent variable by detecting its pure children from observed variables, which can be achieved under strong graphical and distributional assumptions. We notice that previously used assumptions are sufficient but not necessary for detecting the pure children, and some special impure children can play a similar role as pure ones. Based on this, we formulate two sets of assumptions which are milder than previous ones. They both allow causal edges between observed variables, one allows entirely arbitrary distribution and requires only one pure child per latent variable, and the other requires no pure child and imposes the non-Gaussianity requirement on only a subset of variables, two illustrative examples are shown in Figure 1. We prove identifiability of latent variables under either set of assumptions, and corresponding algorithms directly derive from our constructive proof. After this, we perform some pre-processing procedures and then modify the PC-MIMBuild (Silva et al., 2006) which has already been proved asymptotically correct to infer causal relations between any two variables.

In summary, our main contributions are three-fold. First, we introduce two sets of milder assumptions, both of which avoid the non-Gaussianity, purity, and two-pure-children assumption simultaneously. Second, we prove that the whole causal structure of linear latent variable models can be recovered under either set of assumptions. Third, from our constructive proof, we derive algorithms to recover the whole causal structure from purely observational data, which are both theoretically sound and computationally efficient.

## 2 PRELIMINARIES

In this paper, we focus on linear latent variable models with graph structure $\mathcal{G} = (\mathbf{V}, \mathbf{E})$ which is a directed acyclic graph (DAG). $\mathbf{V} = \mathbf{L} \cup \mathbf{O}$ where $\mathbf{L} = \{L_i\}_i$ and $\mathbf{O} = \{O_i\}_i$ respectively denote the set of latent and observed variables. In the causal graph $\mathcal{G}$, each variable follows:

$$L_i = \sum_{L_j \in \text{Pa}_{\mathbf{L}}^{\mathcal{G}}(L_i)} b_{ji} L_j + \epsilon_{L_i}, \quad O_i = \sum_{L_j \in \text{Pa}_{\mathbf{L}}^{\mathcal{G}}(O_i)} c_{ji} L_j + \sum_{O_j \in \text{Pa}_{\mathbf{O}}^{\mathcal{G}}(O_i)} d_{ji} O_j + \epsilon_{O_i}, \quad (1)$$

where $\text{Pa}_{\mathbf{L}}^{\mathcal{G}}(V), \text{Pa}_{\mathbf{O}}^{\mathcal{G}}(V)$ respectively denote the set of latent parents and observed parents of $V$ in $\mathcal{G}$. Moreover, $\text{Pa}^{\mathcal{G}}(V) = \text{Pa}_{\mathbf{L}}^{\mathcal{G}}(V) \cup \text{Pa}_{\mathbf{O}}^{\mathcal{G}}(V), \text{Pa}^{\mathcal{G}}(\mathbf{V}) = \cup_{V \in \mathbf{V}} \text{Pa}^{\mathcal{G}}(V)$, and $\text{Ch}^{\mathcal{G}}(\cdot), \text{Nei}^{\mathcal{G}}(\cdot)$ respectively denote children and neighbors. $b_{ji}, c_{ji}, d_{ji}$ respectively denote the causal strength from $L_j$ to $L_i$, from $L_j$ to $O_i$, from $O_j$ to $O_i$. $\epsilon_{L_i}$ and $\epsilon_{O_i}$ refer to noises, which are continuous and independent of any other noise. Without loss of generality, we suppose that each variable has zero mean.

**Definition 1.** *(linear latent variable model) A causal model with graph structure $\mathcal{G} = (\mathbf{V}, \mathbf{E})$ where $\mathcal{G}$ is a DAG and $\mathbf{V} = \mathbf{L} \cup \mathbf{O}$ is called a linear latent variable model if*

  *1. each variable follows Equation (1);*

  *2. the distribution over $\mathbf{V}$ is both Markov and faithful to $\mathcal{G}$;*

Equation (1) implies that all causal relations are linear and no observed variable is a parent of any latent one, both of which have almost become standard assumptions of latent causal structure learning since proposed by the seminal work (Silva et al., 2006), although very few works avoid them at the expense of other significant limitations. For instance, Kivva et al. (2021) allow non-linearity, but they assume that all latent variables are discrete. By the way, some of out theoretical results can still generalize to certain special nonlinear cases, please see Appendix C.1 for more details.

**Definition 2.** *(Pure child) An observed variable $O \in \mathbf{O}$ is called a pure child of a latent variable $L \in \mathbf{L}$ if $\mathrm{Pa}^{\mathcal{G}}(O) = \{L\}$ and $\mathrm{Ch}^{\mathcal{G}}(O) = \emptyset$.*[1]

**Example 1.** *In Figure 1(a), $L_1$ has 3 pure children: $O_1, O_2, O_3$; $L_2$ has 2 pure children: $O_4, O_5$; $L_3$ has only 1 pure child $O_{10}$.*

**Definition 3.** *(Pure pair) An observed pair $\{O_1, O_2\} \subset \mathbf{O}$ is called a pure pair if $\exists L \in \mathbf{L}$ s.t. both $O_1$ and $O_2$ are pure children of $L$.*

**Definition 4.** *(Pseudo-pure pair) An observed pair $\{O_1, O_2\} \subset \mathbf{O}$ is called a pseudo-pure pair if $\exists L \in \mathbf{L}$ s.t. (a) $\mathrm{Pa}^{\mathcal{G}}(O_1) = \{L\}, \mathrm{Ch}^{\mathcal{G}}(O_1) = \{O_2\}, \mathrm{Pa}^{\mathcal{G}}(O_2) = \{L, O_1\}, \mathrm{Ch}^{\mathcal{G}}(O_2) = \emptyset$ or (b) $\mathrm{Pa}^{\mathcal{G}}(O_2) = \{L\}, \mathrm{Ch}^{\mathcal{G}}(O_2) = \{O_1\}, \mathrm{Pa}^{\mathcal{G}}(O_1) = \{L, O_2\}$ and $\mathrm{Ch}^{\mathcal{G}}(O_1) = \emptyset$.*

**Definition 5.** *(Generalized pure pair) An observed pair $\{O_1, O_2\} \subset \mathbf{O}$ is called a generalized pure pair if it is either a pure pair or a pseudo-pure pair.*

**Example 2.** *In Figure 1(a), there are 4 pure pairs: $\{O_1, O_2\}, \{O_1, O_3\}, \{O_2, O_3\}, \{O_4, O_5\}$, 3 pseudo-pure pairs: $\{O_6, O_7\}, \{O_8, O_9\}, \{O_{11}, O_{12}\}$, and hence 7 generalized pure pairs.*

Xie et al. (2023b) suggest that with only the two-pure-children assumption, latent variables cannot be fully identified. The problem is that a pseudo-pure pair may be falsely identified as a pure pair, once this happens, a single latent variable will be split into multiple ones, an example is shown in Figure 2. In fact, pseudo-pure pairs are not uncommon in the real world, e.g., in psychometric questionnaires, "insomnia" and "concentration" might be a pseudo-pure pair because the former directly influences

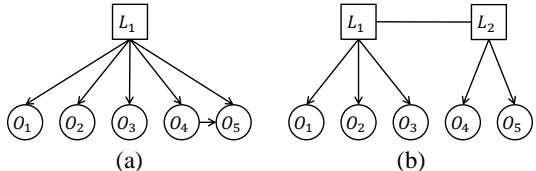

(a)    (b)

Figure 2: With only the two-pure-children assumption, the structure in (a) cannot be discriminated against (b).

the latter and they are also confounded by a latent variable "depression". To handle this problem, they further introduce the non-Gaussianity assumption which enables discrimination between pure and pseudo-pure pairs, while such ambiguity can also be avoided with the previously used three-pure-children assumption (Kummerfeld & Ramsey, 2016). However, both the non-Gaussianity and three-pure-children assumption are only sufficient but not necessary conditions. Besides, we find that pseudo-pure pairs may even benefit causal discovery because they can play a similar role as pure pairs. This motivates us to investigate the case where none of the non-Gaussianity, purity, and two-pure-children assumption holds.

## 3  IDENTIFYING LATENT VARIABLES

To recover the whole causal structure, the first step is to identify latent variables. To this end, we formulate Assumption 1, 2, 3 at the outset of Section 3.1, 3.2, 3.3 respectively. Assumption 1 is a preliminary assumption enabling partial identification of latent variables, which can be achieved by Algorithm 1. On this basis, if Assumption 2 or Assumption 3 is also satisfied, latent variables can be fully identified. Taking the output of Algorithm 1 as the input, Algorithm 2 and 3 can accomplish this goal under Assumption 2 and 3 respectively.

### 3.1  PARTIALLY IDENTIFYING LATENT VARIABLES

**Assumption 1.** *(a) $\forall L \in \mathbf{L}$, L has at least one generalized pure pair as children, (b) $\forall L \in \mathbf{L}$, $\mathrm{Nei}^{\mathcal{G}}_{\mathbf{L}}(L) \neq \emptyset$, and (c) $\forall O \in \mathbf{O}$, if $\mathrm{Pa}^{\mathcal{G}}(O) = \emptyset$, then $|\mathrm{Ch}^{\mathcal{G}}(O)| \geq 3$.*

---

[1] Some recent works such as Xie et al. (2022) focused on the scenario where pure children may still be latent, which is out of our scope. We discuss the relation between these works and ours in Section 5.

---

**Algorithm 1:** Partially identifying latent variables.

---

**Input:** Observed variable $\mathbf{O}$
**Output:** Candidate variables $\mathbf{O}^C$, generalized pure pairs $\mathbb{S}$, purity indicator function $\mathbb{1}_{\text{pure}}(\cdot)$

**1** Find all candidate variables based on Definition 6.
**2** Find all generalized pure pairs based on Theorem 1.
**3** Identify as many pure pairs as possible based on Lemma 1.

---

Assumption 1(a) indicates that a latent variable with a pseudo-pure pair as children can have no pure child. Assumption 1(b) has already been used by previous works like Kummerfeld & Ramsey (2016), which can be replaced by a much weaker assumption, please see Appendix C.2 for more details. Assumption 1(c) means that each root observed variable has a sufficient number of children.

**Definition 6.** *(Candidate variable) Given an observed variable $O_1 \in \mathbf{O}$, we call $O_1$ is a candidate variable if $\forall\{O_i, O_j\} \subset \mathbf{O}\setminus\{O_1\}$, $\exists O_2 \in \mathbf{O}\setminus\{O_1, O_i, O_j\}$ s.t. $O_1 \not\perp\!\!\!\perp O_2$, $O_1 \not\perp\!\!\!\perp O_2|\{O_i\}$, $O_1 \not\perp\!\!\!\perp O_2|\{O_j\}$, and $O_1 \not\perp\!\!\!\perp O_2|\{O_i, O_j\}$.*

We denote the set of candidate variables by $\mathbf{O}^C$.

**Definition 7.** *(Tetrad constraint) Given an observed pair $\{O_1, O_2\} \subset \mathbf{O}$ and a set of observed variables $\mathbf{O}' \subset \mathbf{O}\setminus\{O_1, O_2\}$ s.t. $|\mathbf{O}'| \geq 2$, we call $(\{O_1, O_2\}, \mathbf{O}')$ satisfies the tetrad constraint if $\forall\{O_i, O_j\} \subset \mathbf{O}', \text{Cov}(O_1, O_j)\text{Cov}(O_2, O_i) = \text{Cov}(O_1, O_i)\text{Cov}(O_2, O_j)$.*

**Theorem 1.** *Suppose the underlying linear latent variable model satisfies Assumption 1 and $\{O_1, O_2\} \subset \mathbf{O}$. Then $\{O_1, O_2\} \subset \mathbf{O}^C$ and $(\{O_1, O_2\}, \mathbf{O}\setminus\{O_1, O_2\})$ satisfies the tetrad constraint if and only if $\{O_1, O_2\}$ is a generalized pure pair.*

Based on Theorem 1, we can detect all generalized pure pairs, the set of which is denoted by $\mathbb{S}$. Since each latent variable has at least one generalized pure pair as children according to Assumption 1(a), every latent variable can be detected at least once. The soundness of Theorem 1 heavily relies on Assumption 1. Roughly speaking, Assumption 1(a) and 1(b) guarantees that if $O_1$ and $O_2$ both have latent parents, they are always candidate variables, and the tetrad constraint is satisfied if and only if $\{O_1, O_2\}$ is a generalized pure pair. Assumption 1(c) guarantees that if $O_1$ or $O_2$ has no latent parent, they are not both candidate variables or the tetrad constraint is not satisfied.

**Lemma 1.** *Suppose $\mathcal{S} \in \mathbb{S}$. Then $\mathcal{S}$ is a pure pair if (but not only if) $\exists \mathcal{S}' \in \mathbb{S}$ s.t. $\mathcal{S} \cap \mathcal{S}' \neq \emptyset$.*

To fully identify latent variables, we still need to determine whether any two generalized pure pairs share a common latent parent, which requires us to first discriminate pure pairs against pseudo-pure pairs. Unfortunately, this issue can only be partially addressed based on Lemma 1 at this point, so latent variables can only be partially identified. We define a *purity indicator function* $\mathbb{1}_{\text{pure}}(\cdot)$ on $\mathbb{S}$. If $\mathcal{S} \in \mathbb{S}$ is identified as a pure pair, $\mathbb{1}_{\text{pure}}(\mathcal{S}) = 1$; if $\mathcal{S}$ is identified as a pseudo-pure pair, $\mathbb{1}_{\text{pure}}(\mathcal{S}) = 0$; otherwise, if $\mathcal{S} \in \mathbb{S}$ is unidentifiable temporarily, $\mathbb{1}_{\text{pure}}(\mathcal{S}) = -1$. The algorithm is summarized in Algorithm 1, which has $O(|\mathbf{O}|^4)$ complexity. A detailed version can be found in Appendix D.

## 3.2 FULLY IDENTIFYING LATENT VARIABLES: CASE I

**Assumption 2.** *(a) $\forall L \in \mathbf{L}$, $L$ has at least one pure child, (b) $\forall L \in \mathbf{L}$, $|\text{Nei}^{\mathcal{G}}(L)| \geq 4$. Furthermore, if $|\text{Nei}^{\mathcal{G}}(L)| = 4$, $\text{Nei}_{\mathbf{L}}^{\mathcal{G}}(L) = \{L'\}$, and $\text{Ch}_{\mathbf{O}}^{\mathcal{G}}(L) = \{O_1, O_2, O_3\}$ where $\{O_1, O_2\}$ is a pure pair, then $\text{Nei}^{\mathcal{G}}(O_3) \neq \{L, L'\}$.*

Assumption 2(a) requires only one pure child per latent variable. On the basis of Assumption 1 and 2(a), given an $L \in \mathbf{L}$, there are only two cases where Assumption 2(b) is violated.

1. $L$ has one latent neighbor $L'$, two pure children $\{O_1, O_2\}$, and no other neighbor;
2. $L$ has one latent neighbor $L'$, three observed children $\{O_1, O_2, O_3\}$, and no other neighbor, where $\{O_1, O_2\}$ is a pure pair, $O_3$ has two latent parents $\{L, L'\}$ and no other neighbor.

In other words, Assumption 2(b) can be satisfied in various forms, including but not limited to $|\text{Nei}_{\mathbf{L}}^{\mathcal{G}}(L)| \geq 2$, or $|\text{Nei}_{\mathbf{O}}^{\mathcal{G}}(L)| \geq 4$, or $\exists O \in \text{Ch}_{\mathbf{O}}^{\mathcal{G}}(L)$ s.t. $\text{Nei}_{\mathbf{O}}^{\mathcal{G}}(O) \neq \emptyset$, etc. In particular, if the three-pure-children assumption is satisfied, Assumption 2 holds, so we say our assumption is milder.

**Lemma 2.** *Suppose the underlying linear latent variable model satisfies Assumption 1 and 2, $\mathcal{S} = \{O_1, O_2\} \in \mathbb{S}$ and $\mathbb{1}_{\text{pure}}(\mathcal{S}) = -1$. Then $\mathcal{S}$ is a pseudo-pure pair if and only if $\exists O_3 \in \mathbf{O}^C\setminus\{O_1, O_2\}$ s.t. $(\{O_1, O_3\}, \mathbf{O}\setminus\{O_1, O_2, O_3\})$ satisfies the tetrad constraint.*

---

**Algorithm 2:** Fully identifying latent variables in Case I.

---

**Input:** Observed variables $\mathbf{O}$, candidate variables $\mathbf{O}^C$, generalized pure pairs $\mathbb{S}$, purity indicator function $\mathbb{1}_{\text{pure}}(\cdot)$

**Output:** Updated purity indicator function $\mathbb{1}_{\text{pure}}(\cdot)$, sibling indicator function $\mathbb{1}_{\text{sib}}(\cdot, \cdot)$.

1 Discriminate pure pairs against pseudo-pure pairs based on Lemma 2.

2 Check whether two generalized pure pairs share a common latent parent based on Proposition 1.

---

Given a pseudo-pure pair $\mathcal{S} = \{O_1, O_2\}$, we denote by $\text{Ref}(\mathcal{S})$ any (not all) auxiliary variable $O_3$ that satisfies the condition in Lemma 2.

**Corollary 1.** *Suppose $\mathcal{S}$ is a pseudo-pure pair. Then $\text{Ref}(\mathcal{S})$ is a pure child of $\text{Pa}_{\mathbf{L}}^{\mathcal{G}}(\mathcal{S})$.*

Based on Lemma 2, we can completely discriminate pure pairs against pseudo-pure pairs. Besides, Corollary 1 indicates that $\text{Ref}(\mathcal{S})$ is not an ordinary variable but a pure child of $\text{Pa}_{\mathbf{L}}^{\mathcal{G}}(\mathcal{S})$. The soundness of Lemma 2 heavily relies on Assumption 2. With Assumption 2(a), for any pseudo-pure pair, its latent parent has at least one pure child, which can serve as the auxiliary variable that makes the tetrad constraint in Lemma 2 hold; Without Assumption 2(b), given a pure pair, the tetrad constraint in Lemma 2 may still hold, an example is shown in Figure 2(b): for a pure pair $\{O_4, O_5\}$, $(\{O_4, O_1\}, \mathbf{O}\backslash\{O_1, O_4, O_5\})$ satisfies the tetrad constraint, where $\mathbf{O}\backslash\{O_1, O_4, O_5\} = \{O_2, O_3\}$.

**Proposition 1.** *Let $\{\mathcal{S}_1, \mathcal{S}_2\} \subset \mathbb{S}$.*

1. *Suppose $\mathcal{S}_1$ and $\mathcal{S}_2$ are two pure pairs. Then $\text{Pa}_{\mathbf{L}}^{\mathcal{G}}(\mathcal{S}_1) = \text{Pa}_{\mathbf{L}}^{\mathcal{G}}(\mathcal{S}_2)$ if and only if (1) $\mathcal{S}_1 \cap \mathcal{S}_2 \neq \emptyset$, or (2) $\exists \mathcal{S}_3 \in \mathbb{S}$ s.t. $\mathcal{S}_1 \cap \mathcal{S}_3 \neq \emptyset$ and $\mathcal{S}_2 \cap \mathcal{S}_3 \neq \emptyset$.*

2. *Suppose $\mathcal{S}_1$ is a pure pair and $\mathcal{S}_2$ is a pseudo-pure pair. Then $\text{Pa}_{\mathbf{L}}^{\mathcal{G}}(\mathcal{S}_1) = \text{Pa}_{\mathbf{L}}^{\mathcal{G}}(\mathcal{S}_2)$ if and only if (1) $\text{Ref}(\mathcal{S}_2) \in \mathcal{S}_1$, or (2) $\exists \mathcal{S}_3 \in \mathbb{S}$ s.t. $\text{Ref}(\mathcal{S}_2) \in \mathcal{S}_3$ and $\mathcal{S}_1 \cap \mathcal{S}_3 \neq \emptyset$.*

3. *Suppose $\mathcal{S}_1$ and $\mathcal{S}_2$ are two pseudo-pure pairs. Then $\text{Pa}_{\mathbf{L}}^{\mathcal{G}}(\mathcal{S}_1) = \text{Pa}_{\mathbf{L}}^{\mathcal{G}}(\mathcal{S}_2)$ if and only if (1) $\text{Ref}(\mathcal{S}_1) = \text{Ref}(\mathcal{S}_2)$, or (2) $\exists \mathcal{S}_3 \in \mathbb{S}$ s.t. $\text{Ref}(\mathcal{S}_1) \in \mathcal{S}_3$ and $\text{Ref}(\mathcal{S}_2) \in \mathcal{S}_3$.*

Based on Proposition 1, we can determine whether any two generalized pure pairs share a common latent parent. We define a *sibling indicator function* $\mathbb{1}_{\text{sib}}(\cdot, \cdot)$ on $\mathbb{S} \times \mathbb{S}$. If $\{\mathcal{S}_1, \mathcal{S}_2\} \subset \mathbb{S}$ share a common latent parent, $\mathbb{1}_{\text{sib}}(\mathcal{S}_1, \mathcal{S}_2) = 1$; otherwise, $\mathbb{1}_{\text{sib}}(\mathcal{S}_1, \mathcal{S}_2) = 0$. The algorithm for fully identifying latent variables is summarized in Algorithm 2, which has $O(|\mathbf{O}|^4)$ complexity. A detailed version can be found in Appendix D. With its output, we assign each $\mathcal{S}_i \in \mathbb{S}$ with a latent variable $L_i$, and let $L_i = L_j$ if $\mathbb{1}_{\text{sib}}(\mathcal{S}_i, \mathcal{S}_j) = 1$, such that latent variables are fully identified.

**Theorem 2.** *Suppose the underlying linear latent variable model satisfies Assumption 1 and 2. Then latent variables can be fully identified.*

### 3.3 FULLY IDENTIFYING LATENT VARIABLES: CASE II

**Assumption 3.** *(a) $\forall \mathcal{S}_i = \{O_{i_1}, O_{i_2}\} \in \mathbb{S}$ s.t. $\forall \mathcal{S}_j \in \mathbb{S}\backslash\{\mathcal{S}_i\}, \mathcal{S}_i \cap \mathcal{S}_j = \emptyset$, $\epsilon_{O_{i_1}}$ and $\epsilon_{O_{i_2}}$ are both non-Gaussian. (b) $\forall \mathcal{S} \in \mathbb{S}$ with $\text{Pa}_{\mathbf{L}}^{\mathcal{G}}(\mathcal{S}) = \{L\}$, if $\mathcal{S}$ is a pseudo-pure pair, then $\exists V_1 \in \text{Ch}^{\mathcal{G}}(L)\backslash\mathcal{S}$ s.t. $L \perp\!\!\!\perp \text{Pa}^{\mathcal{G}}(V_1)\backslash\{L\}$. Furthermore, if $\text{Pa}^{\mathcal{G}}(L) = \emptyset$, then $\exists V_2 \in \text{Ch}^{\mathcal{G}}(L)\backslash\mathcal{S}$ s.t. $L \perp\!\!\!\perp \text{Pa}^{\mathcal{G}}(V_2)\backslash\{L\}$ and $V_1 \perp\!\!\!\perp V_2 | L$.*

Assumption 3(a) imposes the non-Gaussianity requirement on only some generalized pure pairs, which are exactly those on which $\mathbb{1}_{\text{pure}}(\cdot)$ is -1. Assumption 3(b) is a bit complicated. Specifically, for any pseudo-pure pair $\mathcal{S}$ with latent parent $L$,

1. if $L$ is a non-root node, Assumption 3(b) requires that $\exists V \in \text{Ch}^{\mathcal{G}}(L)\backslash\mathcal{S}$ s.t. there is no mediator or confounder between $L$ and $V$, where $V$ is not necessarily an observed variable and $V$ may have other parent besides $L$;

2. if $L$ is a root node, Assumption 3(b) requires that $\exists\{V_1, V_2\} \subset \text{Ch}^{\mathcal{G}}(L)\backslash\mathcal{S}$ s.t. there is no mediator between $L$ and $V_1$, no mediator between $L$ and $V_2$, and no confounder between $V_1$ and $V_2$ besides $L$.

Clearly, if the non-Gaussianity assumption holds, Assumption 3(a) is satisfied; if the two-pure-children assumption holds, Assumption 3(b) is satisfied, so we say our assumption is milder. By the way, it is obvious that if Assumption 2 holds, Assumption 3(b) is satisfied.

**Lemma 3.** *Suppose the underlying linear latent variable model satisfies Assumption 1 and 3, $\mathcal{S} = \{O_1, O_2\}$ and $\mathbb{1}_{\text{pure}}(\mathcal{S}) = -1$. Then $\mathcal{S}$ is a pseudo-pure pair if and only if $\exists(O_3, O_4) \subset \mathbf{O}\backslash\{O_1, O_2\}$*

---

**Algorithm 3:** Fully identifying latent variables in Case II.

---

**Input:** Observed variables $\mathbf{O}$, generalized pure pairs $\mathbb{S}$, purity indicator function $\mathbb{1}_{\text{pure}}(\cdot)$
**Output:** Updated indicator function $\mathbb{1}_{\text{pure}}(\cdot)$, sibling indicator function $\mathbb{1}_{\text{sib}}(\cdot, \cdot)$

1  Discriminate pure pairs against pseudo-pure pairs based on Lemma 3.
2  Check whether two generalized pure pairs share a common latent parent based on Proposition 2.

---

*which is an ordered pair s.t. $O_1 + \alpha O_2 + \beta O_3 \perp\!\!\!\perp O_1$ where $\alpha, \beta$ satisfy*

$$\text{Var}(O_1) + \alpha\text{Cov}(O_1, O_2) + \beta\text{Cov}(O_1, O_3) = 0, \tag{2}$$

$$\text{Cov}(O_1, O_4) + \alpha\text{Cov}(O_2, O_4) + \beta\text{Cov}(O_3, O_4) = 0; \tag{3}$$

*or $O_2 + \alpha O_1 + \beta O_3 \perp\!\!\!\perp O_2$ where $\alpha, \beta$ satisfy*

$$\text{Var}(O_2) + \alpha\text{Cov}(O_2, O_1) + \beta\text{Cov}(O_2, O_3) = 0, \tag{4}$$

$$\text{Cov}(O_2, O_4) + \alpha\text{Cov}(O_1, O_4) + \beta\text{Cov}(O_3, O_4) = 0. \tag{5}$$

**Corollary 2.** *Suppose $\mathcal{S} = \{O_1, O_2\} \in \mathbb{S}$ and $\exists(O_3, O_4) \subset \mathbf{O} \backslash \{O_1, O_2\}$ which is an ordered pair s.t. $O_1 + \alpha O_2 + \beta O_3 \perp\!\!\!\perp O_1$ where $\alpha, \beta$ satisfy Equation (2) and (3). Then $\tilde{\mathcal{S}} = \{\tilde{O}_1, \tilde{O}_2\}$ is a pure pair with latent parent $\text{Pa}_{\mathbf{L}}^{\mathcal{G}}(\mathcal{S})$ where $\tilde{O}_1 = O_1$ and $\tilde{O}_2 = O_2 + \frac{1}{\alpha}O_1$.*

Based on Lemma 3, we can completely discriminate pure pairs against pseudo-pure pairs. Besides, each pseudo-pure pair $\mathcal{S}$ can be converted into a pure one $\tilde{\mathcal{S}}$ based on Corollary 2. The soundness of Lemma 3 heavily relies on Assumption 3. Without Assumption 3(a), $O_1 + \alpha O_2 + \beta O_3 \perp\!\!\!\perp O_1$ in Lemma 3 may hold even if $\{O_1, O_2\}$ is a pure pair since if $O_1, O_2, O_3$ are all Gaussian, Equation (2) entails $O_1 + \alpha O_2 + \beta O_3 \perp\!\!\!\perp O_1$. Assumption 3(b) ensures that for any pseudo-pure pair $\mathcal{S}$ with latent parent $L$, we can find two auxiliary variables $\{O_3, O_4\}$ which makes the condition in Lemma 3 hold. If $L$ is not a root node, $O_3$ can be $V_1$ (or its child) in Assumption 3(b), $O_4$ can be an observed child of $L$'s any parent; if $L$ is a root node, $O_3, O_4$ can be $V_1, V_2$ (or their children) in Assumption 3(b).

**Proposition 2.** *Let $\{\mathcal{S}_1, \mathcal{S}_2\} \subset \mathbb{S}$ where $\mathcal{S}_1 = \{O_1, O_2\}$ and $\mathcal{S}_2 = \{O_3, O_4\}$. Then*

1. *Suppose $\mathcal{S}_1$ and $\mathcal{S}_2$ are two pure pairs. Then $\text{Pa}_{\mathbf{L}}^{\mathcal{G}}(\mathcal{S}_1) = \text{Pa}_{\mathbf{L}}^{\mathcal{G}}(\mathcal{S}_2)$ if and only if (1) $\mathcal{S}_1 \cap \mathcal{S}_2 \neq \emptyset$, or (2) $\exists \mathcal{S}_3 \in \mathbb{S}$ s.t. $\mathcal{S}_1 \cap \mathcal{S}_3 \neq \emptyset$ and $\mathcal{S}_2 \cap \mathcal{S}_3 \neq \emptyset$.*

2. *Suppose $\mathcal{S}_1$ is a pure pair and $\mathcal{S}_2$ is a pseudo-pure pair. Then $\text{Pa}_{\mathbf{L}}^{\mathcal{G}}(\mathcal{S}_1) = \text{Pa}_{\mathbf{L}}^{\mathcal{G}}(\mathcal{S}_2)$ if and only if $(\{O_2, \tilde{O}_3\}, \{O_1, \tilde{O}_4\})$ satisfies the tetrad constraint.*

3. *Suppose $\mathcal{S}_1$ and $\mathcal{S}_2$ are two pseudo-pure pairs. Then $\text{Pa}_{\mathbf{L}}^{\mathcal{G}}(\mathcal{S}_1) = \text{Pa}_{\mathbf{L}}^{\mathcal{G}}(\mathcal{S}_2)$ if and only if $(\{\tilde{O}_2, \tilde{O}_3\}, \{\tilde{O}_1, \tilde{O}_4\})$ satisfies the tetrad constraint.*

Based on Proposition 2, we can determine whether two generalized pure pairs share a common latent parent. The algorithm for fully identifying latent variables is summarized in Algorithm 3, which has $O(|\mathbf{O}|^3)$ complexity. A detailed version can be found in Appendix D. With its output, we assign each $\mathcal{S}_i \in \mathbb{S}$ with a latent variable $L_i$, and let $L_i = L_j$ if $\mathbb{1}_{\text{sib}}(\mathcal{S}_i, \mathcal{S}_j) = 1$, such that latent variables are fully identified.

**Theorem 3.** *Suppose the underlying linear latent variable model satisfies Assumption 1 and 3. Then latent variables can be fully identified.*

### 3.4 Discussion

In Section 3.2 and 3.3, we respectively formulate two cases where none of the non-Gaussianity, purity, and two-pure-children assumption holds but latent variable can still be fully identified. These two cases make different trade-offs between graphical and distributional assumption. In terms of the graphical assumption, Case II requiring no pure child is more general than Case I entailing one pure child per latent variable; in terms of the distributional assumption, Case I allowing entirely arbitrary distribution is more general than Case II requiring partial non-Gaussianity.

Algorithm 2 and 3 are proposed to handle Case I and Case II respectively. Without sufficient prior knowledge about the underlying causal model, it is a non-trivial problem to choose between them. We design an expedient to handle this issue. We first run Algorithm 1. After that, if $\forall \mathcal{S} \in \mathbb{S}$, $\mathbb{1}_{\text{pure}}(\mathcal{S}) \neq -1$, latent variables can be fully identified; otherwise, $\forall \{O_i, O_j\} \in \mathbb{S}$ s.t. $\mathbb{1}_{\text{pure}}(\{O_i, O_j\}) = -1$, we find an $O \in \mathbf{O} \backslash \{O_i, O_j\}$ s.t. $\text{Cov}(O_i, O)\text{Cov}(O_j, O) \neq 0$, if one of $O_i, O_j, O_i - \frac{\text{Cov}(O_i, O)}{\text{Cov}(O_j, O)}O_j$ is Gaussian, then $\epsilon_{O_i}$ or $\epsilon_{O_j}$ is Gaussian, violating Assumption 3(a), in

---

**Algorithm 4:** PC-MIMBuild

---

**Input:** Variables $\mathbf{V} = \mathbf{L} \cup \mathbf{O}^D \cup \mathbf{O}^U$, and each variable in $\mathbf{L} \cup \mathbf{O}^U$ has at least two indicators
**Output:** A partially directed acyclic graph $\hat{\mathcal{G}}$ over $\mathbf{V}$

1 Find separation set for variables in $\mathbf{L} \cup \mathbf{O}^U$ based on Theorem 19 in Silva et al. (2006) to recover the skeleton over $\mathbf{L} \cup \mathbf{O}^U$, which is denoted by $\hat{\mathcal{G}}$.
2 Orient v-structures in $\hat{\mathcal{G}}$ based on the separation sets.
3 Orient each undirected edge between a latent variable and an observed variable in $\hat{\mathcal{G}}$.
4 Orient as many undirected edges in $\hat{\mathcal{G}}$ as possible by Meek's rules (Meek, 1995).
5 Add variables in $\mathbf{O}^D$ and corresponding causal edges to $\hat{\mathcal{G}}$.

---

which case we choose Algorithm 2, otherwise we choose Algorithm 3. Although this is not a perfect method, it is much better than making a random choice. In fact, most previous works about latent causal structure learning just assumed some properties of the underlying causal models and circumvented the procedure of testing these assumptions.

## 4 INFERRING CAUSAL RELATIONS BETWEEN ANY TWO VARIABLES

After identifying latent variables, the next step is to infer causal relations between any two variables. In Section 4.1, we describe some pre-processing procedures. In Section 4.2, we present the modified PC-MIMBuild for inferring causal relations. No extra assumption is introduced in this section.

### 4.1 PRE-PROCESSING

Given an observed variable $O$, if $\exists \mathcal{S} \in \mathbb{S}$ s.t. $O \in \mathcal{S}$ in Case I or Case II, or $\exists \mathcal{S} \in \mathbb{S}$ s.t. $O = \text{Ref}(\mathcal{S})$ in Case I, its causal relations with any other variable is determined. Specifically, if $O \in \mathcal{S}$ and $\mathcal{S}$ is a pure pair in Case I or Case II, it has no other neighbor except a latent parent; if $O \in \mathcal{S}$ and $\mathcal{S}$ is a pseudo-pure pair in Case I or Case II, it has no other neighbor except a latent parent and an observed neighbor; if $O = \text{Ref}(\mathcal{S})$ in Case I, it has no other neighbor except a latent parent based on Corollary 1. Such observed variables are called *determined observed variables* otherwise *undetermined observed variables*, the set of which are denoted by $\mathbf{O}^D$ and $\mathbf{O}^U$ respectively. To recover the whole causal structure, we only need to focus on variables in $\mathbf{L} \cup \mathbf{O}^U$.

**Proposition 3.** *No variable in $\mathbf{O}^D$ is a parent of any variable in $\mathbf{L} \cup \mathbf{O}^U$.*

To recover the causal structure over latent variables, PC-MIMBuild requires that each latent variable has at least two measured indicators that can be represented as its linear function plus an independent noise. Given a latent variable $L$, if $L$ has multiple pure children, these pure children can be detected and serve as the indicators of $L$, otherwise, $L$ must has a pseudo-pure pair $\{O_1, O_2\}$ as children according to Assumption 1(a). In Case I, $O_1$ and $\text{Ref}(\{O_1, O_2\})$ can serve as indicators of $L$; in Case II, $\{\tilde{O}_1, \tilde{O}_2\}$ derived by Corollary 2 can serve as indicators of $L$. Furthermore, our objective is to recover the whole causal structure involving both latent and observed variables. Since $\mathbf{L} \cup \mathbf{O}^U$ is causally sufficient based on Proposition 3, if we create two auxiliary indicators for each undetermined observed variable by adding independent noises to it, causal relations between any two variables in $\mathbf{L} \cup \mathbf{O}^U$ can be revealed by PC-MIMBuild.

### 4.2 PC-MIMBUILD

An overview of PC-MIMBuild are summarized in Algorithm 4. Since no observed variable is a parent of any latent one in linear latent variable models, when searching for the separation set of any two latent variables in line 1, we limit the search space to $\mathbf{L}$ to reduce computational cost. For the same reason, we can orient each undirected edge between a latent variable and an observed one in line 3, which allows more undirected edges to be oriented by Meek's rules (Meek, 1995) in line 4.

**Theorem 4.** *Suppose the underlying linear latent variable model satisfies Assumption 1 and 2 or Assumption 1 and 3, in the limit of infinite data, $\hat{\mathcal{G}}$ satisfies that (1) $\hat{\mathcal{G}}$ has the same skeleton and v-structures as $\mathcal{G}$; (2) $\forall \{O_i, O_j\} \subset \mathbf{O}$ s.t. $O_i \in \text{Pa}^{\mathcal{G}}(O_j)$ and $\text{Pa}^{\mathcal{G}}_{\mathbf{L}}(O_i) \neq \text{Pa}^{\mathcal{G}}_{\mathbf{L}}(O_j)$, $O_i \in \text{Pa}^{\hat{\mathcal{G}}}(O_j)$.*

It is not surprising that without further assumptions, the causal structure can only be identified up to Markov equivalence. Fortunately, Chen et al. (2022) have proposed additional distributional

conditions under which the causal structure can be identified completely and also a corresponding algorithm, which could apply to $\hat{\mathcal{G}}$ directly if their proposed conditions hold.

## 5 RELATION TO EXISTING WORK

Most traditional causal discovery approaches typically assume the absence of latent variables, but they usually yield unreliable results in situations involving latent variables which may cause spurious correlations. This has inspired extensive researches into causal discovery with latent variables. While some works aim to reveal causal relations between observed variables in the presence of latent variables, others attempt to identify latent variables and recover the latent causal structure. An important line of works employs sparsity of causal edges to facilitate latent causal structure learning for linear latent variable models. The seminal work (Silva et al., 2006) suggested that the latent causal structure can be identified under the three-pure-children assumption. On this basis, Kummerfeld & Ramsey (2016) proposed a more efficient algorithm which allows partial non-linearity based on the work of Spirtes (2013). Cai et al. (2019) first showed that two-pure-children assumption could also enable identification of latent causal structure. Subsequently, Xie et al. (2020) and Zeng et al. (2021) attempted to generalize the results of Cai et al. (2019) to more challenging scenarios. The former could address the scenario where observed variables have multiple latent parents while the latter could recover the latent causal structure shared by multiple domains. Although the requirement for pure children has been relaxed, they additionally entailed the non-Gaussianity and purity assumption. Xie et al. (2023b) made a further step by eliminating the purity assumption. Instead, we formulate two more general cases where none of the non-Gaussianity, purity, and two-pure-children assumption holds. By the way, the purity assumption is also required by some other works which utilized matrix decomposition (Anandkumar et al., 2013) or mixture oracle (Kivva et al., 2021) for latent causal structure learning, so our work is also more general than theirs in this regard.

Recently, latent hierarchical causal structure learning has drawn significant attention, where "hierarchical" means that some latent variables may lack observed children. The seminal work (Xie et al., 2022) relied on the non-Gaussianity, purity, and generalized two-pure-children (pure children could be either latent or observed) assumption. Huang et al. (2022) used more general assumptions that allow arbitrary distribution. Chen et al. (2023) highlighted that the assumptions in Huang et al. (2022) may not hold if there exist three mutually adjacent variables, they overcome this limitation by requiring more pure children. At a high level, each of these works decomposes the hierarchical structure into multiple layers, and then infers latent variables and their causal relations from lower to higher levels recursively. Since most algorithms used within a single level originate from those designed for conventional linear latent variable models, our results can be potentially generalized to this scenario. Some recent works leveraged counterfactual data (Brehmer et al., 2022; Ahuja et al., 2022) or interventional data (Ahuja et al., 2023; Seigal et al., 2023) rather than purely observational data for latent causal structure learning. Although they have avoided many distributional and graphical assumptions, interventional or counterfactual data is not always available in practice.

Instead of the causal structure over only latent variables, we attempt to recover the whole causal structure involving both latent and observed variables. Previously, Adams et al. (2021) have introduced much weaker graphical assumptions for recovery of the whole causal structure, but they still required the non-Gaussianity assumption and their proposed algorithms are computationally intractable. A contemporaneous work (Xie et al., 2023a) proposed efficient algorithms to recover the whole causal structure of n-factor causal models with latent hierarchical structure, which are beyond our ability. However, they still required the non-Gaussianity and (generalized) two-pure-children assumption, and only allowed edges between particular observed variables.

In this paper, we decompose the recovery of the whole causal structure into two sub-problems: identification of latent variables and inference of causal relations between any two variables. In Section 3, we provide main theoretical results about the first sub-problem, the proofs of which heavily rely on the Tetrad Representation Theorem (Spirtes et al., 2000) and Darmois-Skitovich Theorem (Kagan, 1989), which are presented in Appendix B.1 and B.7. The former builds a connection between the structure of underlying causal model and the covariance of variables that can be calculated from samples. The latter means that as long as two variables share any non-Gaussian component, they cannot be statistically independent. Although many existing works (Silva et al., 2006; Xie et al., 2020; 2023b) also used them as cornerstones, their results cannot be directly involved in our framework where their required assumptions are mostly not satisfied. In Section 4, we

Table 1: Performance on synthetic data derived by causal models with structure $\mathcal{G}_1$ and $\mathcal{G}_2$.

| Graph | Algorithm | N=500 | | | | N=1000 | | | | N=2000 | | | |
|---|---|---|---|---|---|---|---|---|---|---|---|---|---|
| | | LO | LC | WI | Err | LO | LC | WI | Err | LO | LC | WI | Err |
| $\mathcal{G}_1$ | BPC | 0.33±0.37 | 0.00±0.00 | 0.19±0.16 | 1.0 | 0.30±0.28 | 0.00±0.00 | 0.16±0.13 | 1.0 | 0.27±0.13 | 0.00±0.00 | 0.16±0.08 | 1.0 |
| | FOFC | 0.93±0.13 | 0.00±0.00 | 0.02±0.05 | 1.0 | 1.00±0.00 | 0.00±0.00 | 0.00±0.00 | 1.0 | 1.00±0.00 | 0.00±0.00 | 0.00±0.00 | 1.0 |
| | GIN | 0.50±0.27 | 0.00±0.00 | 0.66±0.17 | 1.0 | 0.60±0.20 | 0.00±0.00 | 0.75±0.09 | 1.0 | 0.57±0.21 | 0.00±0.00 | 0.71±0.16 | 1.0 |
| | Ours | 0.07±0.13 | 0.03±0.10 | 0.03±0.06 | **0.4** | 0.03±0.10 | 0.00±0.00 | 0.01±0.04 | **0.2** | 0.00±0.00 | 0.00±0.00 | 0.00±0.00 | **0.0** |
| $\mathcal{G}_2$ | BPC | 0.30±0.23 | 0.00±0.00 | 0.29±0.13 | 1.0 | 0.27±0.20 | 0.00±0.00 | 0.30±0.11 | 1.0 | 0.27±0.13 | 0.00±0.00 | 0.28±0.11 | 1.0 |
| | FOFC | 0.90±0.15 | 0.00±0.00 | 0.03±0.04 | 1.0 | 0.97±0.10 | 0.00±0.00 | 0.01±0.02 | 1.0 | 1.00±0.00 | 0.00±0.00 | 0.00±0.00 | 1.0 |
| | GIN | 0.10±0.15 | 0.07±0.13 | 0.20±0.15 | 1.0 | 0.07±0.13 | 0.07±0.13 | 0.23±0.13 | 1.0 | 0.00±0.00 | 0.07±0.13 | 0.30±0.07 | 1.0 |
| | Ours | 0.13±0.22 | 0.00±0.00 | 0.11±0.17 | **0.5** | 0.10±0.21 | 0.00±0.00 | 0.07±0.15 | **0.3** | 0.03±0.10 | 0.00±0.00 | 0.03±0.10 | **0.2** |

present algorithms for the second sub-problem, which are mostly based on the PC-MIMBuild (Silva et al., 2006). To make it more adaptable to our framework, we design pre-processing procedures in Section 4.1 and also make some modifications to itself in Section 4.2.

# 6 EXPERIMENTAL RESULTS

We apply our proposed algorithms to both synthetic and real-world data to demonstrate their effectiveness. Due to the space limit, we only present experimental results on synthetic data derived by causal models with structure $\mathcal{G}_1$ and $\mathcal{G}_2$ as shown in Figure 1 in the main text and provide more details in Appendix A. For each graph, we draw 10 sample sets of size $N$=500, 1000, 2000 respectively. Each causal strength is sampled from a uniform distribution over $[-2.0, -0.5] \cup [0.5, 2.0]$. Noises of causal models with structure $\mathcal{G}_1$ are Gaussian variables with mean 0 and standard error drawn from uniform(0.5,1), those of causal models with structure $\mathcal{G}_2$ are the seventh power of uniform(-1,1) variables, which are then normalized to have a standard error also drawn from uniform(0.5,1).

We compare our proposed methods with BPC (Silva et al., 2006), FOFC (Kummerfeld & Ramsey, 2016), and GIN (Xie et al., 2020). We use Latent Omission (LO), Latent Commission (LC), Wrong Indicator (WI) as the evaluation metrics. LO and LC are respectively the number of omitted and redundant latent variables divided by the total number of latent variables in ground truth graph. WI is the number of wrong indicators divided by the total number of observed variables in the ground truth graph, where an indicator is called wrong if it measures at least one wrong latent variable or it is still dependent of some other indicators given the latent variable it measures. Besides, we also report the Error Rate (Err) which is the number of sample sets on which LO, LC, and WI are not all 0 divided by the total number of sample sets.

The experimental results are summarized in Table 1. No previous approach can yield correct results since their required assumptions are not satisfied. For instance, FOFC requires the three-pure-children assumption, and its implementation in TETRAD (Ramsey et al., 2018) actually prefers at least four pure children per latent variable, so it cannot detect latent variables, leading to high LO. Using the expedient proposed in Section 3.4, we choose Algorithm 1 plus 2 to recover $\mathcal{G}_1$ and Algorithm 1 plus 3 to recover $\mathcal{G}_2$. Because causal models with structure $\mathcal{G}_1$ and $\mathcal{G}_2$ both satisfy Assumption 1 and respectively satisfy Assumption 2 and 3, our algorithms can return correct results.

# 7 CONCLUSION AND FUTURE WORK

In this paper, we endeavor to recover the whole causal structure of linear latent variable models under milder graphical and distributional assumptions. Firstly, we formulate two cases where none of the non-Gaussianity, purity, and two-pure-children assumption holds. Secondly, we prove that the whole causal structure involving both latent and observed variables is identifiable in either case. Thirdly, we also provide efficient algorithms for causal structure recovery.

Although we prove identifiability under milder assumptions, they may still not hold in practice. For instance, an observed variable might be the cause of some latent variables (Adams et al., 2021), some causal relations might be non-linear (Kaltenpoth & Vreeken, 2023) or non-stationary (Liu & Kuang, 2023), and the underlying causal graph may be cyclic because feedback loops are not uncommon (Sethuraman et al., 2023). Actually, some causal questions might be answered even without a fully identified causal graph, so it is useful to investigate to which extent the causal structure can be recovered under less restrictive assumptions. Finally, to guarantee a trustworthy result, we need special algorithms to test whether the required assumptions are satisfied, for which we only propose an imperfect expedient in Section 3.4 while most existing works directly circumvent this procedure.

ACKNOWLEDGEMENT

XL is partially supported by the JD Technology Scholarship for Postgraduate Research in Artificial Intelligence. KZ would like to acknowledge the support from NSF Grant 2229881, the National Institutes of Health (NIH) under Contract R01HL159805, and grants from Apple Inc., KDDI Research Inc., Quris AI, and Infinite Brain Technology. TL is partially supported by the following Australian Research Council projects: FT220100318, DP220102121, LP220100527, LP220200949, IC190100031.

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

Organization of Appendix:

- Section A: More experimental results.
- Section B: Proof.
- Section C: More theoretical results.
- Section D: Details of algorithms.

## A    MORE EXPERIMENTAL RESULTS

First, we compare Algorithm 1 plus 2 to existing algorithms on more causal models with structure $\mathcal{G}_3, \mathcal{G}_4, \mathcal{G}_5$ as shown in Figure 3(a,b,c). For each graph, we draw 10 sample sets of size $N$=1000. Each causal strength is sampled from a uniform distribution over $[-2.0, -0.5] \cup [0.5, 2.0]$. All noises are Gaussian variables generated following Section 6. Experimental results are summarized in Table 2.

Table 2: Comparison of Algorithm 1 plus 2 with others.

| Algorithm | $\mathcal{G}_3$ | | | | $\mathcal{G}_4$ | | | | $\mathcal{G}_5$ | | | |
|---|---|---|---|---|---|---|---|---|---|---|---|---|
| | LO | LC | WI | Err | LO | LC | WI | Err | LO | LC | WI | Err |
| BPC | 0.00±0.00 | 0.00±0.00 | 0.05±0.09 | 0.2 | 0.67±0.30 | 0.00±0.00 | 0.30±0.26 | 1.0 | 0.40±0.13 | 0.00±0.00 | 0.39±0.12 | 1.0 |
| FOFC | 0.07±0.13 | 0.00±0.00 | 0.01±0.02 | 0.2 | 1.00±0.00 | 0.00±0.00 | 0.00±0.00 | 1.0 | 0.97±0.10 | 0.00±0.00 | 0.02±0.07 | 1.0 |
| GIN | 0.53±0.16 | 0.00±0.00 | 0.58±0.14 | 1.0 | 0.50±0.22 | 0.00±0.00 | 0.78±0.00 | 1.0 | 0.67±0.00 | 0.00±0.00 | 0.78±0.00 | 1.0 |
| Ours | 0.03±0.10 | 0.00±0.00 | 0.00±0.00 | 0.1 | 0.00±0.00 | 0.23±0.15 | 0.40±0.13 | 1.0 | 0.00±0.00 | 0.00±0.00 | 0.13±0.07 | 1.0 |

1. Since $\mathcal{G}_3$ satisfies the three-pure-children assumption, BPC and FOFC can yield correct results. Because Assumption 1 and 2 are also satisfied in this case, our algorithm can also return correct results. As all noises are Gaussian and there exist edges between observed variables in $\mathcal{G}_3$, GIN always produce wrong results.

2. Since $L_1$ in $\mathcal{G}_4$ has no pure child, Assumption 2(a) is not satisfied. In this case, our algorithm will falsely identify both $\{O_1, O_2\}$ and $\{O_3, O_4\}$ as pure pairs which do not share a common latent parent. Therefore, $L_1$ will be split into two latent variables, whose indicators are respectively $\{O_1, O_2\}$ and $\{O_3, O_4\}$, leading to high LC and WI. Therefore, our algorithms cannot yield correct results. Any other algorithm also fails.

3. Since $L_3$ in $\mathcal{G}_5$ has exactly one latent parent $L_2$ and three observed children $O_7, O_8, O_9$ where $O_7, O_8$ are both pure child of $L_3$ and $\mathrm{Pa}^{\mathcal{G}}(O_7) = \{L_2, L_3\}$, Assumption 2(b) is not satisfied. In this case our algorithm will falsely identify $\{O_8, O_9\}$ as a pseudo-pure pair and $\mathrm{Ref}(\{O_8, O_9\}) = O_7$, so $O_7, O_8$ will serve as the indicators of $L_3$, leading to high WI. Therefore, our algorithms cannot yield correct results. Any other algorithm also fails.

Second, we compare Algorithm 1 plus 3 to existing algorithms on more causal models with structure $\mathcal{G}_6, \mathcal{G}_7, \mathcal{G}_8$ as shown in Figure 3(d,e,f). For each graph, we draw 10 sample sets of size $N$=1000. Each causal strength is sampled from a uniform distribution over $[-2.0, -0.5] \cup [0.5, 2.0]$. All noises of causal models with structure $\mathcal{G}_6$ and $\mathcal{G}_8$ are non-Gaussian variables generated following Section 6. For causal models with structure $\mathcal{G}_7$, noises of $\{O_2, O_4, O_6, O_8\}$ are non-Gaussian while other noises are Gaussian. Experimental results are summarized in Table 3.

Table 3: Comparison of Algorithm 1 plus 3 with others.

| Algorithm | $\mathcal{G}_6$ | | | | $\mathcal{G}_7$ | | | | $\mathcal{G}_8$ | | | |
|---|---|---|---|---|---|---|---|---|---|---|---|---|
| | LO | LC | WI | Err | LO | LC | WI | Err | LO | LC | WI | Err |
| BPC | 0.63±0.31 | 0.00±0.00 | 0.14±0.13 | 1.0 | 0.53±0.16 | 0.00±0.00 | 0.35±0.12 | 1.0 | 0.77±0.15 | 0.00±0.00 | 0.28±0.21 | 1.0 |
| FOFC | 1.00±0.00 | 0.00±0.00 | 0.00±0.00 | 1.0 | 0.97±0.10 | 0.00±0.00 | 0.03±0.08 | 1.0 | 1.00±0.00 | 0.00±0.00 | 0.00±0.00 | 1.0 |
| GIN | 0.47±0.43 | 0.00±0.00 | 0.00±0.00 | 0.6 | 0.50±0.27 | 0.03±0.10 | 0.68±0.11 | 1.0 | 0.53±0.22 | 0.03±0.10 | 0.75±0.00 | 1.0 |
| Ours | 0.00±0.00 | 0.00±0.00 | 0.00±0.00 | 0.0 | 0.43±0.21 | 0.00±0.00 | 0.65±0.12 | 1.0 | 0.07±0.20 | 0.00±0.00 | 0.30±0.10 | 1.0 |

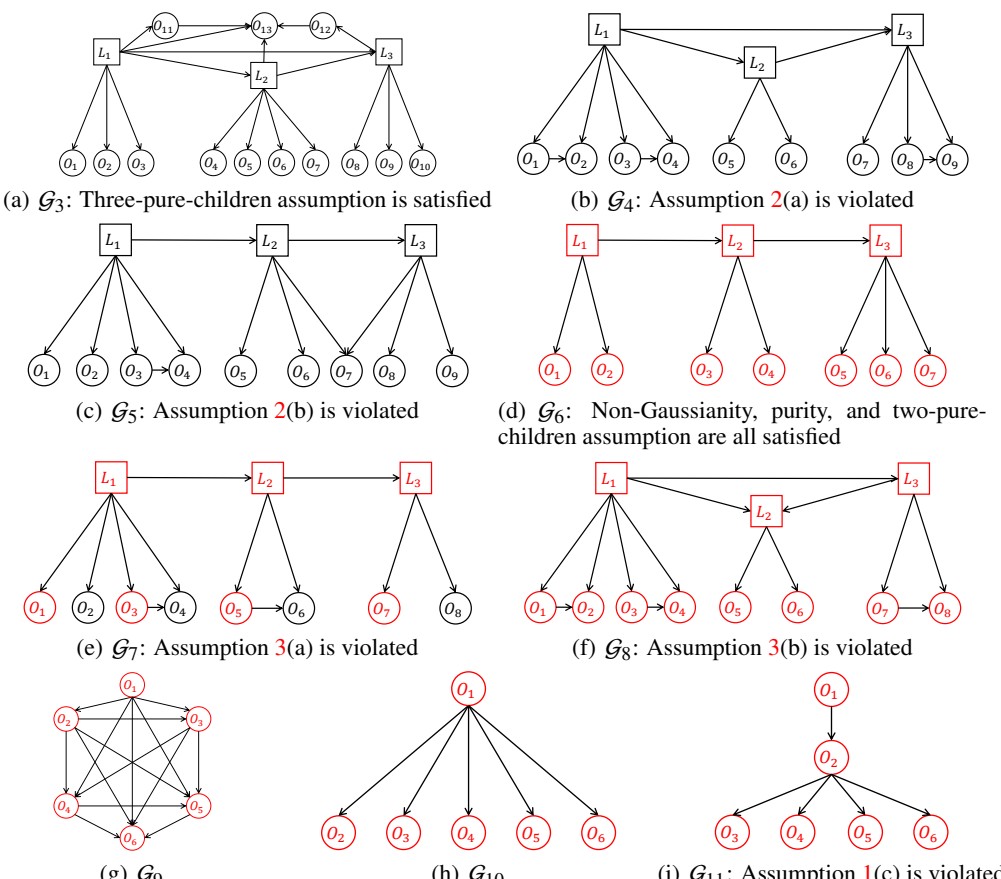

Figure 3: Causal graphs used to synthesize data, noises of red variables are non-Gaussian.

1. Since causal models with structure $\mathcal{G}_6$ satisfy the non-Gaussianity, purity, and two-pure-children assumption, GIN can yield correct results. In this case, Assumption 1 and 3 are also satisfied, so our algorithms can also return correct results. Furthermore, because algorithms use both second-order and high-order statistics while GIN purely relies on high-order statistics, our Err is remarkably lower than GIN. Since the three-pure-children assumption is not satisfied, both BPC and FOFC always produce wrong results.

2. Since $\mathbb{S} = \{\{O_1, O_2\}, \{O_3, O_4\}, \{O_5, O_6\}, \{O_7, O_8\}\}$ in $\mathcal{G}_7$ and noises of $\{O_2, O_4, O_6, O_8\}$ in causal models with structure $\mathcal{G}_7$ are all Gaussian, Assumption 3(a) is not satisfied. In this case, our algorithm may falsely identify pure pairs as pseudo-pure ones and pseudo-pure pairs cannot be converted into pure ones correctly. It is possible that a single latent variable is split into multiple ones or multiple latent variables are merged into a signle one. Therefore, our algorithms cannot yield correct results. Any other algorithm also fails.

3. Since the $L_3$ in $\mathcal{G}_8$ has a pseudo-pure pair $\{O_7, O_8\}$ as children but has no other child, Assumption 3(b) is not satisfied. In this case, our algorithm will falsely identify $\{O_7, O_8\}$ as a pure one, and both of them will serve as the indicators of $L_3$, leading to high WI. Therefore, our algorithms cannot yield correct results. Any other algorithm also fails.

Moreover, we investigate the behavior of different algorithms on causal models with structure $\mathcal{G}_9$, $\mathcal{G}_{10}$, and $\mathcal{G}_{11}$ as shown in Figure 3(g,h,i), where there is no latent variable. For each graph, we draw 10 sample sets of size $N$=1000. Each causal strength is sampled from a uniform distribution over $[-2.0, -0.5] \cup [0.5, 2.0]$. All noises are non-Gaussian variables generated following Section 6. Overall, our algorithms return no generalized pure pair in $\mathcal{G}_9, \mathcal{G}_{10}$, and $\mathcal{G}_{11}$, indicating that there exists no latent variable. This is because no tetrad constraint is satisfied in $\mathcal{G}_9$ and there exists only one candidate variable $O_1$ and $O_2$ in $\mathcal{G}_{10}$ and $\mathcal{G}_{11}$ respectively. However, all other algorithms introduce a latent variable being the parent of all observed ones in $\mathcal{G}_{10}$ and $\mathcal{G}_{11}$. Note that in $\mathcal{G}_{10}$, the

root variable $O_1$ has only one child $O_2$, violating Assumption 1(c), but our algorithms can still yield correct results. In the case without latent variable, PC-MIMBuild can be replaced by the vanilla PC.

Finally, we compare our algorithms with others on two real-world datasets: HolzingerSwineford1939 (HS1939) (Rosseel, 2012) and Teacher Burnout (TB) (Byrne, 2013). The experimental results are summarized in Table 4.

Table 4: Experimental results on real-world datasets.

| Algorithm | HS1939 | | | Teacher Burnout | | |
|---|---|---|---|---|---|---|
| | LO | LC | WI | LO | LC | WI |
| BPC | 0.00 | 0.00 | 0.00 | 0.42 | 0.00 | 0.13 |
| FOFC | 1.00 | 0.00 | 0.00 | 1.00 | 0.00 | 0.00 |
| GIN | 0.33 | 0.00 | 0.22 | 0.75 | 0.00 | 0.34 |
| Ours | 0.00 | 0.00 | 0.00 | 0.25 | 0.00 | 0.06 |

1. HolzingerSwineford1939 (HS1939) dataset consists of mental ability test scores of seventh- and eighth-grade children. 9 observed variables are influenced by 3 latent variables (visual, textual and speed). Please refer to Section 4 in Rosseel (2012) for more details of the dataset and the ground-truth causal graph.

2. Teacher Burnout (TB) dataset is used to investigate the influence of organization and personality on burnout in full-time elementary teachers. 32 observed variables are influenced by 12 latent variables (role ambiguity, role conflict, work overload, classroom climate, decision making, superior support, peer support, self-esteem, external locus of control, emotional exhaustion, depersonalization and personal accomplishment). Please refer to Chapter 6 in Byrne (2013) for more details of the dataset and the ground-truth causal graph.

## B  PROOFS

### B.1  PROOF OF THEOREM 1

We begin with the Tetrad Representation Theorem (Spirtes et al., 2000) which is essential for our proof.

**Theorem 5.** *(Tetrad Representation Theorem) In a linear latent variable model with graph structure $\mathcal{G}$, let $I_1, I_2, J_1, J_2$ be four variables in $\mathcal{G}$. Then $(\{I_1, I_2\}, \{J_1, J_2\})$ satisfies the tetrad constraint if and only if there is a choke point between $\{I_1, I_2\}$ and $\{J_1, J_2\}$.*

Theorem 5 entails some graphical definitions. Specifically, in a directed acyclic (DAG) graph,

1. a *path* is a sequence of distinct variables $V_1, ..., V_m$, such that $V_k$ is adjacent to $V_{k+1}$ for all $k = 1, ..., m - 1$. In particular, a single variable is also a path.

2. a *collider* on a path $V_1, ..., V_m$ is a variable $V_i$, $1 < i < n$, such that $V_{i-1}$ and $V_{i+1}$ are both parents of $V_i$;

3. a *trek* is a path that does not contain any collider;

4. the *source* of a trek is the unique node in the trek to which no arrows are directed.

5. the *I side* of a trek between $I$ and $J$ with source $S$ is the subpath directed from $S$ to $I$. In particular, it is possible that $S = I$;

6. a *choke point* between two set of variables $\mathbf{I}$ and $\mathbf{J}$ is a variable $V$ that lies on every trek between any element of $\mathbf{I}$ and any element of $\mathbf{J}$, and $V$ is either on the $\mathbf{I}$ side of every such trek or on the $\mathbf{J}$ side of every such trek.

Then we further define *pseudo-pure child* and *generalized pure child* for ease of exposition.

**Definition 8.** *(Pseudo-pure child) An observed variable $O_1 \in \mathbf{O}$ is called a pseudo-pure child of a latent variable $L \in \mathbf{L}$ if $O_1 \in \text{Ch}^{\mathcal{G}}_{\mathbf{O}}(L)$ and $\exists O_2$ s.t. $\{O_1, O_2\}$ is a pseudo-pure pair. Furthermore, if $O_1 \in \text{Pa}^{\mathcal{G}}(O_2)$, $O_1$ is called a type-I pseudo-pure child of $L$; otherwise, $O_1$ is called a type-II pseudo-pure child of $L$.*

**Definition 9.** *(Generalized pure child) An observed variable $O \in \mathbf{O}$ is called a generalized pure child of a latent variable $L \in \mathbf{L}$ if $O$ is a pure child or a pseudo-pure child of $L$.*

**Assumption 1.** *(a) $\forall L \in \mathbf{L}$, $L$ has at least one generalized pure pair as children, (b) $\forall L \in \mathbf{L}$, $\mathrm{Nei}_{\mathbf{L}}^{\mathcal{G}}(L) \neq \emptyset$, and (c) $\forall O \in \mathbf{O}$, if $\mathrm{Pa}^{\mathcal{G}}(O) = \emptyset$, then $|\mathrm{Ch}^{\mathcal{G}}(O)| \geq 3$.*

Assumption 1(a) indicates that each latent variable has at least **two** generalized pure children.

Before proving Theorem 1, we present three lemmas about candidate variables under Assumption 1.

**Lemma 4.** *Suppose $O_1 \in \mathbf{O}$ and $\mathrm{Pa}_{\mathbf{L}}^{\mathcal{G}}(O_1) \neq \emptyset$. Then $O_1 \in \mathbf{O}^{\mathrm{C}}$.*

*Proof.* Let $L_1 \in \mathrm{Pa}_{\mathbf{L}}^{\mathcal{G}}(O_1)$. According to Assumption 1(a), $L_1$ has a generalized pure child $O_2 \neq O_1$. Besides, $L_1$ has a latent neighbor $L_2$ according to Assumption 1(b) and $L_2$ has two generalized pure children $\{O_3, O_4\} \subset \mathbf{O}\setminus\{O_1, O_2\}$ according to Assumption 1(a). Therefore, there are 3 treks $O_1 \leftarrow L_1 \rightarrow O_2, O_1 \leftarrow L_1 - L_2 \rightarrow O_3, O_1 \leftarrow L_1 - L_2 \rightarrow O_4$. That is, $\forall \{O_i, O_j\} \subset \mathbf{O}\setminus\{O_1\}$, let $O_k \in \{O_2, O_3, O_4\}\setminus\{O_i, O_j\}$, we have $O_1 \not\perp O_k, O_1 \not\perp O_k|\{O_i\}, O_1 \not\perp O_k|\{O_j\}$, and $O_1 \not\perp O_k|\{O_i, O_j\}$, so $O_1 \in \mathbf{O}^{\mathrm{C}}$. □

**Lemma 5.** *Suppose $O_1 \in \mathbf{O}$ and $\mathrm{Pa}_{\mathbf{L}}^{\mathcal{G}}(O_1) = \emptyset$. If $O_1 \in \mathbf{O}^{\mathrm{C}}$, $|\mathrm{Nei}_{\mathbf{O}}^{\mathcal{G}}(O_1)| \geq 2$.*

*Proof.* This lemma is proved by contradiction. Suppose $|\mathrm{Nei}_{\mathbf{O}}^{\mathcal{G}}(O_1)| < 2$, combined with Assumption 1(c), $O_1$ is a leaf node which has no other neighbor besides an observed parent $O_2$. Based on the local Markov property (Peters et al., 2017), $O_1 \perp\!\!\!\perp \mathbf{O}\setminus\{O_1, O_2\}|\{O_2\}$, so $O_1 \notin \mathbf{O}^{\mathrm{C}}$, which leads to contradiction. □

**Lemma 6.** *Suppose $\{O_1, O_2\} \subset \mathbf{O}$ and $\mathrm{Pa}_{\mathbf{L}}^{\mathcal{G}}(O_1) = \mathrm{Pa}_{\mathbf{L}}^{\mathcal{G}}(O_2) = \emptyset$. If $\{O_1, O_2\} \subset \mathbf{O}^{\mathrm{C}}$, then $|\mathrm{Nei}_{\mathbf{O}}^{\mathcal{G}}(O_1) \cup \mathrm{Nei}_{\mathbf{O}}^{\mathcal{G}}(O_2)\setminus\{O_1, O_2\}| \geq 2$.*

*Proof.* This lemma is proved by contradiction. Suppose $|\mathrm{Nei}_{\mathbf{O}}^{\mathcal{G}}(O_1) \cup \mathrm{Nei}_{\mathbf{O}}^{\mathcal{G}}(O_2)\setminus\{O_1, O_2\}| \leq 1$. There are two possible cases.

1. Suppose $O_2 \notin \mathrm{Nei}_{\mathbf{O}}^{\mathcal{G}}(O_1)$, then $|\mathrm{Nei}_{\mathbf{O}}^{\mathcal{G}}(O_1) \cup \mathrm{Nei}_{\mathbf{O}}^{\mathcal{G}}(O_2)\setminus\{O_1, O_2\}| \leq 1$ implies that $|\mathrm{Nei}_{\mathbf{O}}^{\mathcal{G}}(O_1)| \leq 1$, based on Lemma 5, $O_1 \notin \mathbf{O}^{\mathrm{C}}$, which leads to contradiction.

2. Suppose $O_2 \in \mathrm{Nei}_{\mathbf{O}}^{\mathcal{G}}(O_1)$. If $|\mathrm{Nei}_{\mathbf{O}}^{\mathcal{G}}(O_1) \cup \mathrm{Nei}_{\mathbf{O}}^{\mathcal{G}}(O_2)\setminus\{O_1, O_2\}| = 0$, then $\mathrm{Nei}_{\mathbf{O}}^{\mathcal{G}}(O_2) = \{O_1\}$, based on Lemma 5, $O_1 \notin \mathbf{O}^{\mathrm{C}}$, which leads to contradiction. If $|\mathrm{Nei}_{\mathbf{O}}^{\mathcal{G}}(O_1) \cup \mathrm{Nei}_{\mathbf{O}}^{\mathcal{G}}(O_2)\setminus\{O_1, O_2\}| = 1$, since $|\mathrm{Nei}_{\mathbf{O}}^{\mathcal{G}}(O_1)| \geq 2$ and $|\mathrm{Nei}_{\mathbf{O}}^{\mathcal{G}}(O_2)| \geq 2$, there exists $O_3 \in \mathbf{O}\setminus\{O_1, O_2\}$ s.t. $\mathrm{Nei}_{\mathbf{O}}^{\mathcal{G}}(O_1) = \{O_2, O_3\}$ and $\mathrm{Nei}_{\mathbf{O}}^{\mathcal{G}}(O_2) = \{O_1, O_3\}$. According to Assumption 1(c), neither $O_1$ nor $O_2$ is a root node, so either $O_1$ or $O_2$ is a leaf node. Without loss of generality, let $O_1$ be a leaf node, then $O_1$ has no other neighbor besides two observed parents $O_2, O_3$. Based on the local Markov property, $O_1 \perp\!\!\!\perp \mathbf{O}\setminus\{O_2, O_3\}$, that is, $O_1 \notin \mathbf{O}^{\mathrm{C}}$, which leads to contradiction.

□

**Theorem 1.** *Suppose the underlying linear latent variable model satisfies Assumption 1 and $\{O_1, O_2\} \subset \mathbf{O}$. Then $\{O_1, O_2\} \subset \mathbf{O}^{\mathrm{C}}$ and $(\{O_1, O_2\}, \mathbf{O}\setminus\{O_1, O_2\})$ satisfies the tetrad constraint if and only if $\{O_1, O_2\}$ is a generalized pure pair.*

*Proof.* (i) "If". Suppose $\{O_1, O_2\}$ is a generalized pure pair, then $\mathrm{Pa}_{\mathbf{L}}^{\mathcal{G}}(O_1) \neq \emptyset$ and $\mathrm{Pa}_{\mathbf{L}}^{\mathcal{G}}(O_2) \neq \emptyset$, based on Lemma 4, $\{O_1, O_2\} \subset \mathbf{O}^{\mathrm{C}}$. Let $\mathrm{Pa}_{\mathbf{L}}^{\mathcal{G}}(\{O_1, O_2\}) = \{L\}$, then $\forall O \in \mathbf{O}\setminus\{O_1, O_2\}$, $L$ lies in each trek between $O_1$ and $O$ and is on the $O_1$ side of every such trek. This also holds for each trek between $O_2$ and $O$. Therefore, $L$ is a choke point between $\{O_1, O_2\}$ and $\mathbf{O}\setminus\{O_1, O_2\}$. Based on Theorem 5, we reach the conclusion that $(\{O_1, O_2\}, \mathbf{O}\setminus\{O_1, O_2\})$ satisfies the tetrad constraint.

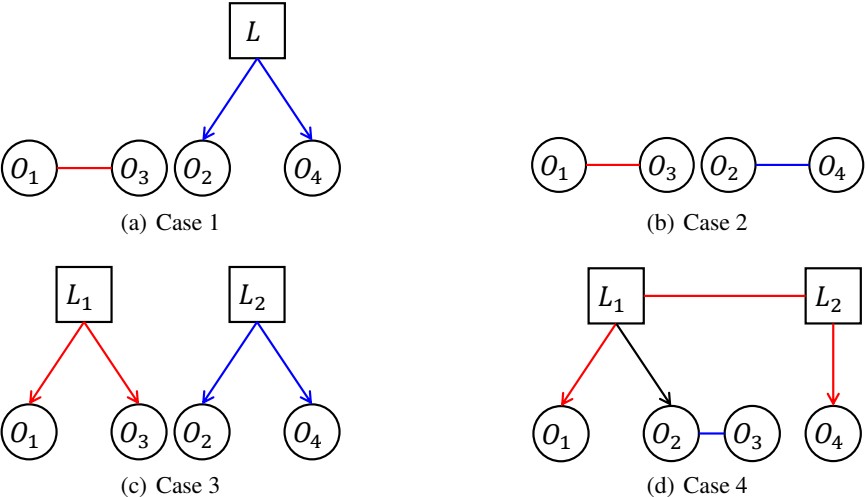

Figure 4: Illustration of proof (ii) of Theorem 1.

(ii) "Only if". This part is proved by contradiction. Suppose $\{O_1, O_2\}$ is not a generalized pure pair, there are 4 possible cases w.r.t. $\mathrm{Pa}_{\mathbf{L}}^{\mathcal{G}}(O_1)$ and $\mathrm{Pa}_{\mathbf{L}}^{\mathcal{G}}(O_2)$.

1. Suppose ones of $\mathrm{Pa}_{\mathbf{L}}^{\mathcal{G}}(O_1)$ and $\mathrm{Pa}_{\mathbf{L}}^{\mathcal{G}}(O_2)$ is an empty set. Without loss of generality, we suppose $\mathrm{Pa}_{\mathbf{L}}^{\mathcal{G}}(O_1) = \emptyset$ and $\mathrm{Pa}_{\mathbf{L}}^{\mathcal{G}}(O_2) \neq \emptyset$. Based on Lemma 5, $O_1$ has an observed neighbor $O_3 \in \mathbf{O} \backslash \{O_2\}$. Let $L \in \mathrm{Pa}_{\mathbf{L}}^{\mathcal{G}}(O_2)$, then $L$ has a generalized pure child $O_4 \in \mathbf{O} \backslash \{O_1, O_2, O_3\}$ according to Assumption 1(a). Clearly, there are two non-intersecting treks $O_1 - O_3$ and $O_2 \leftarrow L \rightarrow O_4$. Therefore, there is no choke point between $\{O_1, O_2\}$ and $\{O_3, O_4\}$. An illustrative example is shown in Figure 4(a).

2. Suppose $\mathrm{Pa}_{\mathbf{L}}^{\mathcal{G}}(O_1) = \emptyset$ and $\mathrm{Pa}_{\mathbf{L}}^{\mathcal{G}}(O_2) = \emptyset$. Based on Lemma 5 and 6, $\exists \{O_3, O_4\} \in \mathbf{O} \backslash \{O_1, O_2\}$ s.t. $O_3 \in \mathrm{Nei}_{\mathbf{O}}^{\mathcal{G}}(O_1)$ and $O_4 \in \mathrm{Nei}_{\mathbf{O}}^{\mathcal{G}}(O_2)$. Clearly, there are two non-intersecting treks $O_1 - O_3$ and $O_2 - O_4$. Therefore, there is no choke point between $\{O_1, O_2\}$ and $\{O_3, O_4\}$. An illustrative example is shown in Figure 4(b).

3. Suppose $\mathrm{Pa}_{\mathbf{L}}^{\mathcal{G}}(O_1) \neq \emptyset, \mathrm{Pa}_{\mathbf{L}}^{\mathcal{G}}(O_2) \neq \emptyset$ and $|\mathrm{Pa}_{\mathbf{L}}^{\mathcal{G}}(\{O_1, O_2\})| > 1$. That is, there exist $L_1 \in \mathrm{Pa}_{\mathbf{L}}^{\mathcal{G}}(O_1), L_2 \in \mathrm{Pa}_{\mathbf{L}}^{\mathcal{G}}(O_2)$ s.t. $L_1 \neq L_2$. According to Assumption 1(a), $L_1$ has a generalized pure child $O_3 \in \mathbf{O} \backslash \{O_1, O_2\}$. Similarly, $L_2$ has a generalized pure child $O_4 \in \mathbf{O} \backslash \{O_1, O_2\}$ and $O_3 \neq O_4$. Clearly, there are two non-intersecting treks $O_1 \leftarrow L_1 \rightarrow O_3$ and $O_2 \leftarrow L_2 \rightarrow O_4$. Therefore, there is no choke point between $\{O_1, O_2\}$ and $\{O_3, O_4\}$. An illustrative example is shown in Figure 4(c).

4. Suppose $\mathrm{Pa}_{\mathbf{L}}^{\mathcal{G}}(O_1) \neq \emptyset, \mathrm{Pa}_{\mathbf{L}}^{\mathcal{G}}(O_2) \neq \emptyset$ and $|\mathrm{Pa}_{\mathbf{L}}^{\mathcal{G}}(\{O_1, O_2\})| = 1$. Since $\{O_1, O_2\}$ is not a generalized pure pair, there is $\mathrm{Nei}_{\mathbf{O}}^{\mathcal{G}}(O_1) \backslash \{O_2\} \neq \emptyset$ or $\mathrm{Nei}_{\mathbf{O}}^{\mathcal{G}}(O_2) \backslash \{O_1\} \neq \emptyset$. Without loss of generality, we suppose $\mathrm{Pa}_{\mathbf{L}}^{\mathcal{G}}(\{O_1, O_2\}) = \{L_1\}, O_3 \in \mathrm{Nei}_{\mathbf{O}}^{\mathcal{G}}(O_2) \backslash \{O_1\}$. Then $L_1$ has a latent neighbor $L_2$ according to Assumption 1(b) and $L_2$ has a generalized pure child $O_4 \in \mathbf{O} \backslash \{O_1, O_2, O_3\}$ according to Assumption 1(a). Clearly, there are two non-intersecting treks $O_2 - O_3$ and $O_1 \leftarrow L_1 - L_2 \rightarrow O_4$. Therefore, there is no choke point between $\{O_1, O_2\}$ and $\{O_3, O_4\}$. An illustrative example is shown in Figure 4(d).

Based on Theorem 5, we reach the conclusion that $(\{O_1, O_2\}, \mathbf{O} \backslash \{O_1, O_2\})$ **does not** satisfy the tetrad constraint, which leads to contradiction. □

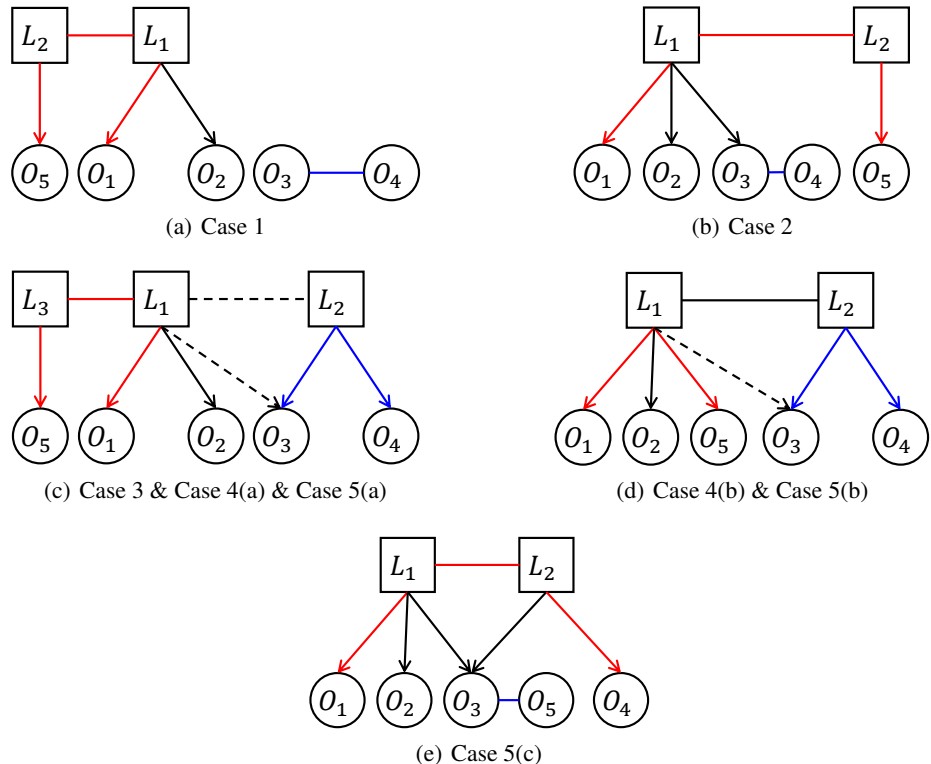

(a) Case 1

(b) Case 2

(c) Case 3 & Case 4(a) & Case 5(a)

(d) Case 4(b) & Case 5(b)

(e) Case 5(c)

Figure 5: Illustration of proof (ii) of Lemma 2, where dotted lines mean that the edges may or may not exist.

## B.2 PROOF OF LEMMA 1

**Lemma 1.** *Suppose $\mathcal{S} \in \mathbb{S}$. Then $\mathcal{S}$ is a pure pair if (but not only if) $\exists \mathcal{S}' \in \mathbb{S}$ s.t. $\mathcal{S} \cap \mathcal{S}' \neq \emptyset$.*

*Proof.* (i) "If". This can be easily derived from the Definition 3, 4, and 5.

(ii) "Not only if". If a latent variable $L$ has no pure child except $O_1$ and $O_2$, then $\{O_1, O_2\}$ is a pure pair but it does not overlap with any other generalized pure pair. □

## B.3 PROOF OF LEMMA 2

**Assumption 2.** *(a) $\forall L \in \mathbf{L}$, $L$ has at least one pure child, (b) $\forall L \in \mathbf{L}$, $|\mathrm{Nei}^{\mathcal{G}}(L)| \geq 4$. Furthermore, if $|\mathrm{Nei}^{\mathcal{G}}(L)| = 4$, $\mathrm{Nei}_{\mathbf{L}}^{\mathcal{G}}(L) = \{L'\}$, and $\mathrm{Ch}_{\mathbf{O}}^{\mathcal{G}}(L) = \{O_1, O_2, O_3\}$ where $\{O_1, O_2\}$ is a pure pair, then $\mathrm{Nei}^{\mathcal{G}}(O_3) \neq \{L, L'\}$.*

**Lemma 2.** *Suppose the underlying linear latent variable model satisfies Assumption 1 and 2, $\mathcal{S} = \{O_1, O_2\} \in \mathbb{S}$ and $\mathbb{1}_{\mathrm{pure}}(\mathcal{S}) = -1$. Then $\mathcal{S}$ is a pseudo-pure pair if and only if $\exists O_3 \in \mathbf{O}^C \setminus \{O_1, O_2\}$ s.t. $(\{O_1, O_3\}, \mathbf{O} \setminus \{O_1, O_2, O_3\})$ satisfies the tetrad constraint.*

*Proof.* (i) "Only if". Suppose $\mathcal{S} = \{O_1, O_2\}$ is a pseudo-pure pair and let $\mathrm{Pa}_{\mathbf{L}}^{\mathcal{G}}(\mathcal{S}) = \{L\}$. According to Assumption 2(a), $L$ has a pure child $O_3 \in \mathbf{O} \setminus \{O_1, O_2\}$. Based on Lemma 4, $O_3 \in \mathbf{O}^C$. Besides, $\forall O \in \mathbf{O} \setminus \{O_1, O_2, O_3\}$, $L$ lies in each trek between $O_1$ and $O$ and is on the $O_1$ side of every such trek. This also holds for each trek between $O_3$ and $O$. Therefore, $L$ is a choke point between $\{O_1, O_3\}$ and $\mathbf{O} \setminus \{O_1, O_2, O_3\}$. Based on Theorem 5, we reach the conclusion that $(\{O_1, O_3\}, \mathbf{O} \setminus \{O_1, O_2, O_3\})$ satisfies the tetrad constraint.

(ii) "If". This part is proved by contradiction. Suppose $\mathcal{S} = \{O_1, O_2\}$ is a pure pair and let $\mathrm{Pa}_{\mathbf{L}}^{\mathcal{G}}(\mathcal{S}) = \{L_1\}$. Because $\mathbb{1}_{\mathrm{pure}}(\mathcal{S}) = -1$, $L_1$ has no pure child except $O_1$ and $O_2$, otherwise we can derive $\mathbb{1}_{\mathrm{pure}}(\mathcal{S}) = 1$ based on Lemma 1. There are 5 possible cases w.r.t. $\mathrm{Pa}_{\mathbf{L}}^{\mathcal{G}}(O_3)$.

1. Suppose $\mathrm{Pa}_{\mathbf{L}}^{\mathcal{G}}(O_3) = \emptyset$. Based on Lemma 5, $O_3$ has an observed children $O_4 \in \mathbf{O} \setminus \{O_1, O_2\}$. Besides, $L_1$ has a latent neighbor $L_2$ according to Assumption 1(b) and $L_2$ has a generalized pure child $O_5 \in \mathbf{O} \setminus \{O_1, O_2, O_3, O_4\}$ according to Assumption 1(a). Clearly, there are two non-intersecting treks $O_3 - O_4$ and $O_1 \leftarrow L_1 - L_2 \rightarrow O_5$. Therefore, there is no choke point between $\{O_1, O_3\}$ and $\{O_4, O_5\}$. An illustrative example is shown in Figure 5(a).

2. Suppose $\mathrm{Pa}_{\mathbf{L}}^{\mathcal{G}}(O_3) = \{L_1\}$. Since $O_3$ is not a pure child of $L_1$, $O_3$ has an observed neighbor $O_4 \in \mathbf{O} \setminus \{O_1, O_2\}$. Besides, $L_1$ has a latent neighbor $L_2$ according to Assumption 1(b) and $L_2$ has a generalized pure child $O_5 \in \mathbf{O} \setminus \{O_1, O_2, O_3, O_4\}$ according to Assumption 1(a). Clearly, there are two non-intersecting treks $O_3 - O_4$ and $O_1 \leftarrow L_1 - L_2 \rightarrow O_5$. Therefore, there is no choke point between $\{O_1, O_3\}$ and $\{O_4, O_5\}$. An illustrative example is shown in Figure 5(b).

3. Suppose $\mathrm{Pa}_{\mathbf{L}}^{\mathcal{G}}(O_3) \neq \{L_1\}$ and $\exists L_2 \in \mathrm{Pa}_{\mathbf{L}}^{\mathcal{G}}(O_3) \setminus \{L_1\}$ s.t. $L_2 \notin \mathrm{Nei}_{\mathbf{L}}^{\mathcal{G}}(L_1)$. Then $L_2$ has a generalized pure child $O_4 \in \mathbf{O} \setminus \{O_1, O_2, O_3\}$ according to Assumption 1(a). Besides, $L_1$ has a latent neighbor $L_3 \neq L_2$ according to Assumption 1(b) and $L_3$ has a generalized pure child $O_5 \in \mathbf{O} \setminus \{O_1, O_2, O_3, O_4\}$ according to Assumption 1(a). Clearly, there are two non-intersecting treks $O_3 \leftarrow L_2 \rightarrow O_4$ and $O_1 \leftarrow L_1 - L_3 \rightarrow O_5$. Therefore, there is no choke point between $\{O_1, O_3\}$ and $\{O_4, O_5\}$. An illustrative example is shown in Figure 5(c).

4. Suppose $\mathrm{Pa}_{\mathbf{L}}^{\mathcal{G}}(O_3) \neq \{L_1\}$, $\forall L \in \mathrm{Pa}_{\mathbf{L}}^{\mathcal{G}}(O_3) \setminus \{L_1\}$, $L \in \mathrm{Nei}_{\mathbf{L}}^{\mathcal{G}}(L_1)$ and $L_1 \notin \mathrm{Pa}_{\mathbf{L}}^{\mathcal{G}}(O_3)$. Let $L_2 \in \mathrm{Pa}_{\mathbf{L}}^{\mathcal{G}}(O_3) \setminus \{L_1\}$, then $L_2$ has a generalized pure child $O_4 \in \mathbf{O} \setminus \{O_1, O_2, O_3\}$ according to Assumption 1(a). Furthermore, there are two possible sub-cases w.r.t. $\mathrm{Nei}_{\mathbf{L}}^{\mathcal{G}}(L_1)$.

   (a) Suppose $|\mathrm{Nei}_{\mathbf{L}}^{\mathcal{G}}(L_1)| \geq 2$. Then $L_1$ has a latent neighbor $L_3 \neq L_2$. The remaining proof is the same as Case 3.

   (b) Suppose $|\mathrm{Nei}_{\mathbf{L}}^{\mathcal{G}}(L_1)| = 1$. According to Assumption 2(b) that $|\mathrm{Nei}^{\mathcal{G}}(L_1)| \geq 4$, $\exists O_5 \in \mathrm{Ch}_{\mathbf{O}}^{\mathcal{G}}(L_1) \setminus \{O_1, O_2\}$. Since $L_1 \notin \mathrm{Pa}_{\mathbf{L}}^{\mathcal{G}}(O_3)$ and $O_4$ is a generalized pure child of $L_2$, we have $O_5 \neq O_3$ and $O_5 \neq O_4$. Clearly, there are two non-intersecting treks $O_3 \leftarrow L_2 \rightarrow O_4$ and $O_1 \leftarrow L_1 \rightarrow O_5$. Therefore, there is no choke point between $\{O_1, O_3\}$ and $\{O_4, O_5\}$. An illustrative example is shown in Figure 5(d).

   According to Assumption 1(b), $\mathrm{Nei}_{\mathbf{L}}^{\mathcal{G}}(L_1) \neq \emptyset$, so there is no other possible sub-case.

5. Suppose $\mathrm{Pa}_{\mathbf{L}}^{\mathcal{G}}(O_3) \neq \{L_1\}$, $\forall L \in \mathrm{Pa}_{\mathbf{L}}^{\mathcal{G}}(O_3) \setminus \{L_1\}$, $L \in \mathrm{Nei}_{\mathbf{L}}^{\mathcal{G}}(L_1)$ and $L_1 \in \mathrm{Pa}_{\mathbf{L}}^{\mathcal{G}}(O_3)$. Let $L_2 \in \mathrm{Pa}_{\mathbf{L}}^{\mathcal{G}}(O_3) \setminus \{L_1\}$, then $L_2$ has a generalized pure child $O_4 \in \mathbf{O} \setminus \{O_1, O_2, O_3\}$ according to Assumption 1(a). Furthermore, there are three possible sub-cases w.r.t. $\mathrm{Nei}_{\mathbf{L}}^{\mathcal{G}}(L_1)$.

   (a) Suppose $|\mathrm{Nei}_{\mathbf{L}}^{\mathcal{G}}(L_1)| \geq 2$. The remaining proof is the same as Case 4(a).

   (b) Suppose $|\mathrm{Nei}_{\mathbf{L}}^{\mathcal{G}}(L_1)| = 1$ and $|\mathrm{Nei}^{\mathcal{G}}(L_1)| > 4$, let $O_5 \in \mathrm{Ch}_{\mathbf{O}}^{\mathcal{G}}(L_1) \setminus \{O_1, O_2, O_3\}$. Since $O_4$ is a generalized pure child of $L_2$, $O_5 \neq O_4$. The remaining proof is the same as Case 4(b).

   (c) Suppose $|\mathrm{Nei}_{\mathbf{L}}^{\mathcal{G}}(L_1)| = 1$ and $|\mathrm{Nei}^{\mathcal{G}}(L_1)| = 4$, considering the supposition at the beginning of Case 5, we have $\mathrm{Nei}_{\mathbf{L}}^{\mathcal{G}}(L_1) = \{L_2\}$, $\mathrm{Ch}_{\mathbf{O}}^{\mathcal{G}}(L_1) = \{O_1, O_2, O_3\}$, $\mathrm{Pa}_{\mathbf{L}}^{\mathcal{G}}(O_3) = \{L_1, L_2\}$. According to Assumption 2(b) that $\mathrm{Nei}^{\mathcal{G}}(O_3) \neq \{L_1, L_2\}$, there is $\mathrm{Nei}_{\mathbf{O}}^{\mathcal{G}}(O_3) \neq \emptyset$. Let $O_5 \in \mathrm{Nei}_{\mathbf{O}}^{\mathcal{G}}(O_3)$. Since $O_4$ is a generalized pure child of $L_2$, $O_5 \neq O_4$. Clearly, there are two non-intersecting treks $O_3 - O_5$ and $O_1 \leftarrow L_1 - L_2 \rightarrow O_4$. Therefore, there is no choke point between $\{O_1, O_3\}$ and $\{O_4, O_5\}$. An illustrative example is shown in Figure 5(e).

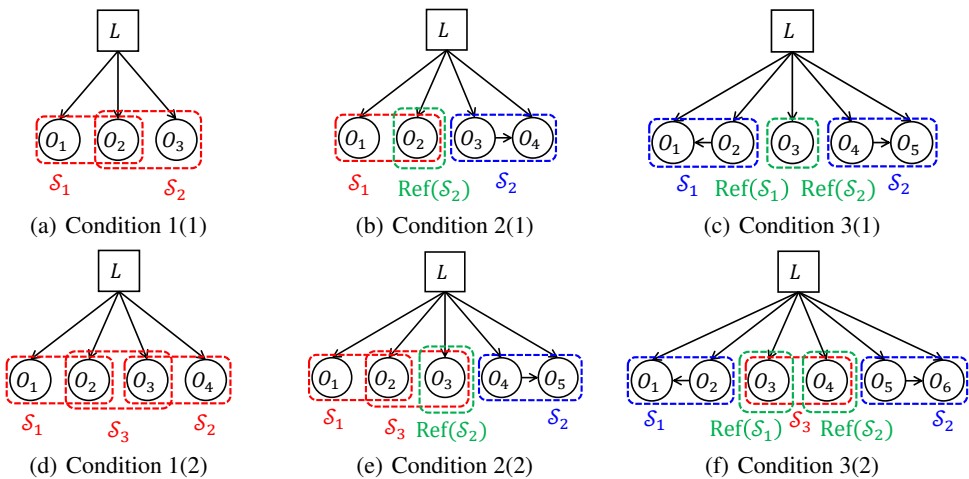

Figure 6: Illustrative examples for Proposition 1.

According to Assumption 1(b) and Assumption 2(b), $\text{Nei}_{\mathbf{L}}^{\mathcal{G}}(L_1) \neq \emptyset$ and $|\text{Nei}^{\mathcal{G}}(L_1)| \geq 4$, there is no other possible sub-case.

Based on Theorem 5, we reach the conclusion that $\forall O_3 \in \mathbf{O}^C \backslash \{O_1, O_2\}$, $(\{O_1, O_3\}, \mathbf{O} \backslash \{O_1, O_2, O_3\})$ **does not** satisfy the tetrad constraint, which leads to contradiction.

□

## B.4    PROOF OF COROLLARY 1

**Corollary 1.** *Suppose $\mathcal{S}$ is a pseudo-pure pair. Then $\text{Ref}(\mathcal{S})$ is a pure child of $\text{Pa}_{\mathbf{L}}^{\mathcal{G}}(\mathcal{S})$.*

*Proof.* This corollary is proved by contradiction. Suppose $\text{Ref}(\mathcal{S})$ is not a pure child of $\text{Pa}_{\mathbf{L}}^{\mathcal{G}}(\mathcal{S})$, let $\mathcal{S} = \{O_1, O_2\}, \text{Ref}(\mathcal{S}) = O_3$ and $\text{Pa}_{\mathbf{L}}^{\mathcal{G}}(\mathcal{S}) = \{L_1\}$. According to Assumption 2(a), $L_1$ has a pure child $O_4 \in \mathbf{O} \backslash \{O_1, O_2, O_3\}$. Then there are three possible cases w.r.t. $\text{Pa}_{\mathbf{L}}^{\mathcal{G}}(O_3)$.

1. Suppose $\text{Pa}_{\mathbf{L}}^{\mathcal{G}}(O_3) = \emptyset$. Based on Lemma 2 and Lemma 5, $O_3$ has an observed neighbor $O_5 \in \mathbf{O} \backslash \{O_1, O_2, O_4\}$. Clearly, there are two non-intersecting treks $O_1 \leftarrow L_1 \rightarrow O_4$ and $O_3 - O_5$. Therefore, there is no choke point between $\{O_1, O_3\}$ and $\{O_4, O_5\}$.

2. Suppose $\text{Pa}_{\mathbf{L}}^{\mathcal{G}}(O_3) \neq \emptyset$ and $\text{Pa}_{\mathbf{L}}^{\mathcal{G}}(O_3) \neq \{L_1\}$. Let $L_2 \in \text{Pa}_{\mathbf{L}}^{\mathcal{G}}(O_3) \backslash \{L_1\}$. Then $L_2$ has a generalized pure child $O_5 \in \mathbf{O} \backslash \{O_1, O_2, O_3, O_4\}$. Clearly, there are two non-intersecting treks $O_1 \leftarrow L_1 \rightarrow O_4$ and $O_3 \leftarrow L_2 \rightarrow O_5$. Therefore, there is no choke point between $\{O_1, O_3\}$ and $\{O_4, O_5\}$.

3. Suppose $\text{Pa}_{\mathbf{L}}^{\mathcal{G}}(O_3) = \{L_1\}$. Since $O_3$ is not a pure child of $L_1$, it has an observed neighbor $O_5 \in \mathbf{O} \backslash \{O_1, O_2, O_3, O_4\}$. Clearly, there are two non-intersecting treks $O_1 \leftarrow L_1 \rightarrow O_4$ and $O_3 - O_5$. Therefore, there is no choke point between $\{O_1, O_3\}$ and $\{O_4, O_5\}$.

Based on Theorem 5, we reach the conclusion that $(\{O_1, O_3\}, \mathbf{O} \backslash \{O_1, O_2, O_3\})$ **does not** satisfy the tetrad constraint, which leads to contradiction. □

## B.5    PROOF OF PROPOSITION 1

**Proposition 1.** *Let $\{\mathcal{S}_1, \mathcal{S}_2\} \subset \mathbb{S}$.*

1. *Suppose $\mathcal{S}_1$ and $\mathcal{S}_2$ are two pure pairs. Then $\mathrm{Pa}_{\mathbf{L}}^{\mathcal{G}}(\mathcal{S}_1) = \mathrm{Pa}_{\mathbf{L}}^{\mathcal{G}}(\mathcal{S}_2)$ if and only if (1) $\mathcal{S}_1 \cap \mathcal{S}_2 \neq \emptyset$, or (2) $\exists \mathcal{S}_3 \in \mathbb{S}$ s.t. $\mathcal{S}_1 \cap \mathcal{S}_3 \neq \emptyset$ and $\mathcal{S}_2 \cap \mathcal{S}_3 \neq \emptyset$.*

2. *Suppose $\mathcal{S}_1$ is a pure pair and $\mathcal{S}_2$ is a pseudo-pure pair. Then $\mathrm{Pa}_{\mathbf{L}}^{\mathcal{G}}(\mathcal{S}_1) = \mathrm{Pa}_{\mathbf{L}}^{\mathcal{G}}(\mathcal{S}_2)$ if and only if (1) $\mathrm{Ref}(\mathcal{S}_2) \in \mathcal{S}_1$, or (2) $\exists \mathcal{S}_3 \in \mathbb{S}$ s.t. $\mathrm{Ref}(\mathcal{S}_2) \in \mathcal{S}_3$ and $\mathcal{S}_1 \cap \mathcal{S}_3 \neq \emptyset$.*

3. *Suppose $\mathcal{S}_1$ and $\mathcal{S}_2$ are two pseudo-pure pairs. Then $\mathrm{Pa}_{\mathbf{L}}^{\mathcal{G}}(\mathcal{S}_1) = \mathrm{Pa}_{\mathbf{L}}^{\mathcal{G}}(\mathcal{S}_2)$ if and only if (1) $\mathrm{Ref}(\mathcal{S}_1) = \mathrm{Ref}(\mathcal{S}_2)$, or (2) $\exists \mathcal{S}_3 \in \mathbb{S}$ s.t. $\mathrm{Ref}(\mathcal{S}_1) \in \mathcal{S}_3$ and $\mathrm{Ref}(\mathcal{S}_2) \in \mathcal{S}_3$.*

*Proof.* (i) "If".

1. Suppose $\mathcal{S}_1$ and $\mathcal{S}_2$ are two pure pairs. (1) Suppose $\mathcal{S}_1 \cap \mathcal{S}_2 \neq \emptyset$. Then $\mathrm{Pa}_{\mathbf{L}}^{\mathcal{G}}(\mathcal{S}_1) = \mathrm{Pa}_{\mathbf{L}}^{\mathcal{G}}(\mathcal{S}_1 \cap \mathcal{S}_2) = \mathrm{Pa}_{\mathbf{L}}^{\mathcal{G}}(\mathcal{S}_2)$. (2) Suppose $\exists \mathcal{S}_3 \in \mathbb{S}$ s.t. $\mathcal{S}_1 \cap \mathcal{S}_3 \neq \emptyset$ and $\mathcal{S}_2 \cap \mathcal{S}_3 \neq \emptyset$. Then $\mathrm{Pa}_{\mathbf{L}}^{\mathcal{G}}(\mathcal{S}_1) = \mathrm{Pa}_{\mathbf{L}}^{\mathcal{G}}(\mathcal{S}_3) = \mathrm{Pa}_{\mathbf{L}}^{\mathcal{G}}(\mathcal{S}_2)$.

2. Suppose $\mathcal{S}_1$ is a pure pair and $\mathcal{S}_2$ is a pseudo-pure pair. (1) Suppose $\mathrm{Ref}(\mathcal{S}_2) \in \mathcal{S}_1$. Then based on Corollary 1, $\mathrm{Pa}_{\mathbf{L}}^{\mathcal{G}}(\mathcal{S}_1) = \mathrm{Pa}_{\mathbf{L}}^{\mathcal{G}}(\mathrm{Ref}(\mathcal{S}_2)) = \mathrm{Pa}_{\mathbf{L}}^{\mathcal{G}}(\mathcal{S}_2)$. (2) Suppose $\exists \mathcal{S}_3 \in \mathbb{S}$ s.t. $\mathrm{Ref}(\mathcal{S}_2) \in \mathcal{S}_3$ and $\mathcal{S}_1 \cap \mathcal{S}_3 \neq \emptyset$. Then based on Corollary 1, $\mathrm{Pa}_{\mathbf{L}}^{\mathcal{G}}(\mathcal{S}_1) = \mathrm{Pa}_{\mathbf{L}}^{\mathcal{G}}(\mathcal{S}_3) = \mathrm{Pa}_{\mathbf{L}}^{\mathcal{G}}(\mathrm{Ref}(\mathcal{S}_2)) = \mathrm{Pa}_{\mathbf{L}}^{\mathcal{G}}(\mathcal{S}_2)$.

3. Suppose $\mathcal{S}_1$ and $\mathcal{S}_2$ are two pseudo-pure pairs. (1) Suppose $\mathrm{Ref}(\mathcal{S}_1) = \mathrm{Ref}(\mathcal{S}_2)$. Then based on Corollary 1, $\mathrm{Pa}_{\mathbf{L}}^{\mathcal{G}}(\mathcal{S}_1) = \mathrm{Pa}_{\mathbf{L}}^{\mathcal{G}}(\mathrm{Ref}(\mathcal{S}_1)) = \mathrm{Pa}_{\mathbf{L}}^{\mathcal{G}}(\mathrm{Ref}(\mathcal{S}_2)) = \mathrm{Pa}_{\mathbf{L}}^{\mathcal{G}}(\mathcal{S}_2)$. (2) Suppose $\exists \mathcal{S}_3 \in \mathbb{S}$ s.t. $\mathrm{Ref}(\mathcal{S}_1) \in \mathcal{S}_3$ and $\mathrm{Ref}(\mathcal{S}_2) \in \mathcal{S}_3$. Then based on Corollary 1, $\mathrm{Pa}_{\mathbf{L}}^{\mathcal{G}}(\mathcal{S}_1) = \mathrm{Pa}_{\mathbf{L}}^{\mathcal{G}}(\mathrm{Ref}(\mathcal{S}_1)) = \mathrm{Pa}_{\mathbf{L}}^{\mathcal{G}}(\mathcal{S}_3) = \mathrm{Pa}_{\mathbf{L}}^{\mathcal{G}}(\mathrm{Ref}(\mathcal{S}_2)) = \mathrm{Pa}_{\mathbf{L}}^{\mathcal{G}}(\mathcal{S}_2)$.

(ii) "Only if".

1. Suppose $\mathcal{S}_1$ and $\mathcal{S}_2$ are two pure pairs and $\mathrm{Pa}_{\mathbf{L}}^{\mathcal{G}}(\mathcal{S}_1) = \mathrm{Pa}_{\mathbf{L}}^{\mathcal{G}}(\mathcal{S}_2) = \{L\}$. Then $\forall O \in \mathcal{S}_1 \cup \mathcal{S}_2$, $O$ is a pure child of $L$. It is obviously possible that $\mathcal{S}_1 \cap \mathcal{S}_2 \neq \emptyset$; otherwise, $\forall O_1 \in \mathcal{S}_1$ and $\forall O_2 \in \mathcal{S}_2$, $\mathcal{S}_3 = \{O_1, O_2\}$ is also a pure pair, that is, $\mathcal{S}_3 \in \mathbb{S}$.

2. Suppose $\mathcal{S}_1$ is a pure pair, $\mathcal{S}_2$ is a pseudo-pure pair, and $\mathrm{Pa}_{\mathbf{L}}^{\mathcal{G}}(\mathcal{S}_1) = \mathrm{Pa}_{\mathbf{L}}^{\mathcal{G}}(\mathcal{S}_2) = \{L\}$. Then based on Corollary 1, $\forall O \in \mathcal{S}_1 \cup \{\mathrm{Ref}(\mathcal{S}_2)\}$, $O$ is a pure child of $L$. It is obviously possible that $\mathrm{Ref}(\mathcal{S}_2) \in \mathcal{S}_1$; otherwise, $\forall O \in \mathcal{S}_1$, $\mathcal{S}_3 = \{O, \mathrm{Ref}(\mathcal{S}_2)\}$ is also a pure pair, that is, $\mathcal{S}_3 \in \mathbb{S}$.

3. Suppose $\mathcal{S}_1$ and $\mathcal{S}_2$ are two pure pairs and $\mathrm{Pa}_{\mathbf{L}}^{\mathcal{G}}(\mathcal{S}_1) = \mathrm{Pa}_{\mathbf{L}}^{\mathcal{G}}(\mathcal{S}_2) = \{L\}$. Then based on Corollary 1, both $\mathrm{Ref}(\mathcal{S}_1)$ and $\mathrm{Ref}(\mathcal{S}_2)$ are pure children of $L$. It is obviously possible that $\mathrm{Ref}(\mathcal{S}_1) = \mathrm{Ref}(\mathcal{S}_2)$; otherwise $\mathcal{S}_3 = \{\mathrm{Ref}(\mathcal{S}_1), \mathrm{Ref}(\mathcal{S}_2)\}$ is also a pure pair, that is, $\mathcal{S}_3 \in \mathbb{S}$.

$\square$

## B.6 Proof of Theorem 2

**Theorem 2.** *Suppose the underlying linear latent variable model satisfies Assumption 1 and 2. Then latent variables can be fully identified.*

*Proof.* Theorem 1 ensures no latent omission under Assumption 1 while Proposition 1 ensures no latent commission under Assumption 1 and 2. Therefore, latent variables can be fully identified. $\square$

## B.7 Proof of Lemma 3

We begin with Darmois-Skitovitch Theorem (Kagan, 1989) which is essential for our proof.

**Theorem 6.** *(Darmois-Skitovitch Theorem) Suppose two random variables $V_1$ and $V_2$ can be represented as linear combinations of independent random variables $\{e_i\}_i$,*

$$V_1 = \sum_i \alpha_i e_i, \quad V_2 = \sum_i \beta_i e_i. \tag{6}$$

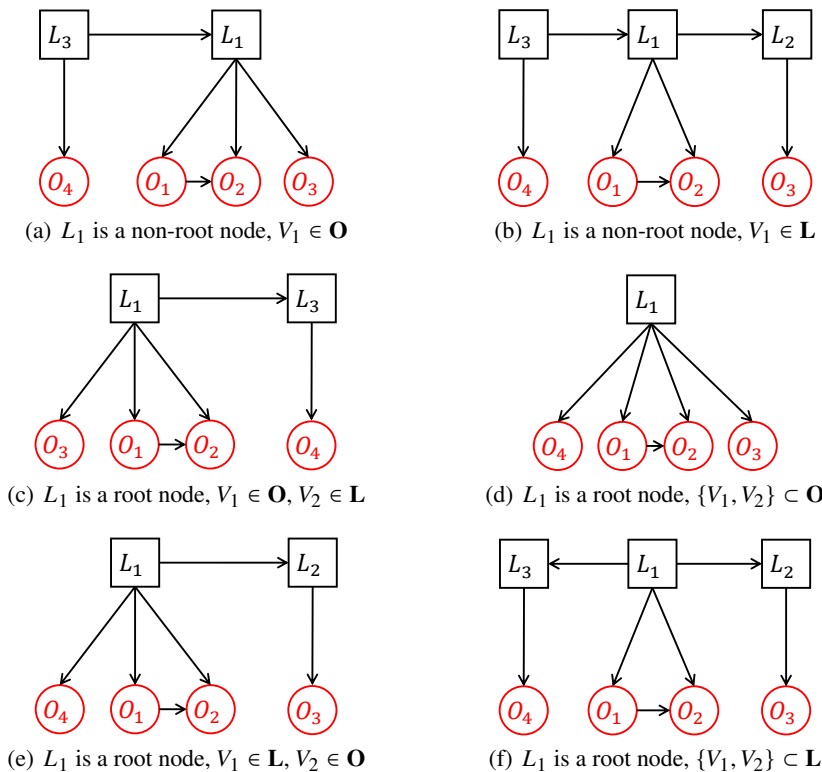

Figure 7: Illustrative of proof of Lemma 3.

(a) $L_1$ is a non-root node, $V_1 \in \mathbf{O}$

(b) $L_1$ is a non-root node, $V_1 \in \mathbf{L}$

(c) $L_1$ is a root node, $V_1 \in \mathbf{O}, V_2 \in \mathbf{L}$

(d) $L_1$ is a root node, $\{V_1, V_2\} \subset \mathbf{O}$

(e) $L_1$ is a root node, $V_1 \in \mathbf{L}, V_2 \in \mathbf{O}$

(f) $L_1$ is a root node, $\{V_1, V_2\} \subset \mathbf{L}$

*Then, if $V_1$ and $V_2$ are independent, all variables $e_j$ for which $\alpha_j \beta_j \neq 0$ are Gaussian. In other words, if there exists a non-Gaussian $e_j$ for which $\alpha_j \beta_j \neq 0$, $V_1$ and $V_2$ are dependent.*

**Assumption 3.** *(a)* $\forall \mathcal{S}_i = \{O_{i_1}, O_{i_2}\} \in \mathbb{S}$ *s.t.* $\forall \mathcal{S}_j \in \mathbb{S} \backslash \{\mathcal{S}_i\}, \mathcal{S}_i \cap \mathcal{S}_j = \emptyset$, $\epsilon_{O_{i_1}}$ *and* $\epsilon_{O_{i_2}}$ *are both non-Gaussian. (b)* $\forall \mathcal{S} \in \mathbb{S}$ *with* $\mathrm{Pa}_{\mathbf{L}}^{\mathcal{G}}(\mathcal{S}) = \{L\}$, *if* $\mathcal{S}$ *is a pseudo-pure pair, then* $\exists V_1 \in \mathrm{Ch}^{\mathcal{G}}(L) \backslash \mathcal{S}$ *s.t.* $L \perp\!\!\!\perp \mathrm{Pa}^{\mathcal{G}}(V_1) \backslash \{L\}$. *Furthermore, if* $\mathrm{Pa}^{\mathcal{G}}(L) = \emptyset$, *then* $\exists V_2 \in \mathrm{Ch}^{\mathcal{G}}(L) \backslash \mathcal{S}$ *s.t.* $L \perp\!\!\!\perp \mathrm{Pa}^{\mathcal{G}}(V_2) \backslash \{L\}$ *and* $V_1 \perp\!\!\!\perp V_2 | L$.

### B.8 Proof of Lemma 3

**Lemma 3.** *Suppose the underlying linear latent variable model satisfies Assumption 1 and 3,* $\mathcal{S} = \{O_1, O_2\}$ *and* $\mathbb{1}_{\mathrm{pure}}(\mathcal{S}) = -1$. *Then* $\mathcal{S}$ *is a pseudo-pure pair if and only if* $\exists (O_3, O_4) \subset \mathbf{O} \backslash \{O_1, O_2\}$ *which is an ordered pair s.t.* $O_1 + \alpha O_2 + \beta O_3 \perp\!\!\!\perp O_1$ *where* $\alpha, \beta$ *satisfy*

$$\mathrm{Var}(O_1) + \alpha \mathrm{Cov}(O_1, O_2) + \beta \mathrm{Cov}(O_1, O_3) = 0, \tag{7}$$

$$\mathrm{Cov}(O_1, O_4) + \alpha \mathrm{Cov}(O_2, O_4) + \beta \mathrm{Cov}(O_3, O_4) = 0, \tag{8}$$

*or* $O_2 + \alpha O_1 + \beta O_3 \perp\!\!\!\perp O_2$ *where* $\alpha, \beta$ *satisfy*

$$\mathrm{Var}(O_2) + \alpha \mathrm{Cov}(O_2, O_1) + \beta \mathrm{Cov}(O_2, O_3) = 0 \tag{9}$$

$$\mathrm{Cov}(O_2, O_4) + \alpha \mathrm{Cov}(O_1, O_4) + \beta \mathrm{Cov}(O_3, O_4) = 0. \tag{10}$$

*Proof.* (i) "If". This part is proved by contradiction. Suppose $\mathcal{S} = \{O_1, O_2\}$ is a pure pair. Then $O_1$ contains $\epsilon_{O_1}$ while $\forall O \notin \mathbf{O} \backslash \{O_1\}$, $O$ does not contain $\epsilon_{O_1}$. Therefore, $\forall \alpha, \beta, O_1 + \alpha O_2 + \beta O_3$ contains $\epsilon_{O_1}$. Because $\epsilon_{O_1}$ is non-Gaussian based on Assumption 3(a), $O_1 + \alpha O_2 + \beta O_3 \not\perp\!\!\!\perp O_1$. Similarly, $\forall \alpha, \beta, O_2 + \alpha O_1 + \beta O_3 \not\perp\!\!\!\perp O_2$, which leads to contradiction.

(ii) "Only if". Suppose $\mathcal{S}$ is a pseudo-pure pair and $O_1 \in \text{Pa}^{\mathcal{G}}(O_2)$ without loss of generality. Let $\text{Pa}_{\mathbf{L}}^{\mathcal{G}}(\{O_1, O_2\}) = \{L_1\}$, then $O_1, O_2$ can be represented as

$$O_1 = c_{11}L_1 + \epsilon_{O_1}, \tag{11}$$

$$O_2 = c_{12}L_1 + d_{12}O_1 + \epsilon_{O_2} = (c_{11}d_{12} + c_{12})L_1 + d_{12}\epsilon_{O_1} + \epsilon_{O_2}. \tag{12}$$

There are two possible cases w.r.t $L_1$.

1. Suppose $L_1$ is not a root node. According to Assumption 3(b), $\exists V_1 \in \text{Ch}^{\mathcal{G}}(L_1)\backslash\{O_1, O_2\}$ s.t. $L_1 \perp\!\!\!\perp \text{Pa}^{\mathcal{G}}(V_1)\backslash\{L_1\}$. If $V_1 \in \mathbf{O}$, we let $O_3 = V_1$; otherwise we let $L_2 = V_1$ and $O_3$ be a pure child or a type-I pseudo-pure child of $L_2$. Therefore, $O_3$ can be represented as

$$O_3 = c_{13}L_1 + \epsilon'_{O_3} \quad \text{or} \quad O_3 = c_{23}L_2 + \epsilon_{O_3} = b_{12}c_{23}L_1 + c_{23}\epsilon'_{L_2} + \epsilon_{O_3}, \tag{13}$$

where $\{\epsilon'_{O_3}, \epsilon'_{L_2}\} \perp\!\!\!\perp L_1$. Let $L_3 \in \text{Pa}_{\mathbf{L}}^{\mathcal{G}}(L_1)$ and $O_4$ be a pure child or a type-I pseudo-pure child of $L_3$ which can be represented as

$$O_4 = c_{34}L_3 + \epsilon_{O_4}. \tag{14}$$

Since $L_1 \perp\!\!\!\perp \text{Pa}^{\mathcal{G}}(V_1)\backslash\{L_1\}$ and $L_3 \in \text{Pa}_{\mathbf{L}}^{\mathcal{G}}(L_1)$, we have $L_3 \perp\!\!\!\perp \text{Pa}^{\mathcal{G}}(V_1)\backslash\{L_1\}$, indicating that $\{\epsilon'_{O_3}, \epsilon'_{L_2}\} \perp\!\!\!\perp L_3$.

2. Suppose $L_1$ is a root node. According to Assumption 3(b), $\exists\{V_1, V_2\} \subset \text{Ch}^{\mathcal{G}}(L)\backslash\mathcal{S}$ s.t. $L \perp\!\!\!\perp \text{Pa}^{\mathcal{G}}(V_1)\backslash\{L\}$, $L \perp\!\!\!\perp \text{Pa}^{\mathcal{G}}(V_2)\backslash\{L\}$ and $V_1 \perp\!\!\!\perp V_2|L$. If $V_1 \in \mathbf{O}$, we let $O_3 = V_1$; otherwise we let $L_2 = V_1$ and $O_3$ be a pure child or a type-I pseudo-pure child of $L_2$. If $V_2 \in \mathbf{O}$, we let $V_2 = O_4$; otherwise we let $V_2 = L_3$ and $O_4$ be a pure child or a type-I pseudo-pure child of $L_3$. Therefore, $O_3, O_4$ can be represented as

$$O_3 = c_{13}L_1 + \epsilon'_{O_3} \quad \text{or} \quad O_3 = c_{23}L_2 + \epsilon_{O_3} = b_{12}c_{23}L_1 + c_{23}\epsilon'_{L_2} + \epsilon_{O_3}, \tag{15}$$

$$O_4 = c_{14}L_1 + \epsilon'_{O_4} \quad \text{or} \quad O_4 = c_{34}L_3 + \epsilon_{O_4} = b_{13}c_{34}L_1 + c_{34}\epsilon'_{L_3} + \epsilon_{O_4}, \tag{16}$$

where $\{\epsilon'_{O_3}, \epsilon'_{L_2}, \epsilon'_{O_4}, \epsilon'_{L_3}\} \perp\!\!\!\perp L_1$ and $\{\epsilon'_{O_3}, \epsilon'_{L_2}\} \perp\!\!\!\perp \{\epsilon'_{O_4}, \epsilon'_{L_3}\}$.

Clearly, if $L_1$ is a non-root node, there are two possible cases w.r.t. $\{O_3, O_4\}$; otherwise, there are four possible cases w.r.t. $\{O_3, O_4\}$. We show all six possible cases in Figure 7. In each case, we can rewrite $O_1, O_2, O_3, O_4$ as

$$O_1 = \lambda_1 L + \epsilon_{O_1}, \tag{17}$$

$$O_2 = \lambda_2 L + \omega_2 \epsilon_{O_1} + \epsilon_{O_2}, \tag{18}$$

$$O_3 = \lambda_3 L + \epsilon''_{O_3}, \tag{19}$$

$$O_4 = \lambda_4 L' + \epsilon''_{O_4}, \tag{20}$$

where $\epsilon_{O_1}, \epsilon_{O_2}, \epsilon''_{O_3}, \epsilon''_{O_4}$ are independent of each other, each of them is independent of $L$ and $L'$, and $\text{Cov}(L, L') \neq 0$. We substitute Equation (17)~(20) into Equation (7) and (8),

$$\lambda_1(\lambda_1 + \alpha\lambda_2 + \beta\lambda_3)\text{Var}(L) + (1 + \alpha\omega_2)\text{Var}(\epsilon_{O_1}) = 0, \tag{21}$$

$$\lambda_4(\lambda_1 + \alpha\lambda_2 + \beta\lambda_3)\text{Cov}(L, L') = 0, \tag{22}$$

which yield that

$$\lambda_1 + \alpha\lambda_2 + \beta\lambda_3 = 0 \quad \text{and} \quad 1 + \alpha\omega_2 = 0. \tag{23}$$

Therefore, we can reach the conclusion that

$$O_1 + \alpha O_2 + \beta O_3 = (\lambda_1 + \alpha\lambda_2 + \beta\lambda_3)L + (1 + \alpha\omega_2)\epsilon_{O_1} + \alpha\epsilon_{O_2} + \beta\epsilon''_{O_3} = \alpha\epsilon_{O_2} + \beta\epsilon''_{O_3} \perp\!\!\!\perp O_1. \tag{24}$$

$\square$

### B.9 PROOF OF COROLLARY 2

**Corollary 2.** *Suppose $\mathcal{S} = \{O_1, O_2\} \in \mathbb{S}$ and $\exists (O_3, O_4) \subset \mathbf{O} \backslash \{O_1, O_2\}$ which is an ordered pair s.t. $O_1 + \alpha O_2 + \beta O_3 \perp\!\!\!\perp O_1$ where $\alpha, \beta$ satisfy Equation (2) and (3). Then $\tilde{\mathcal{S}} = \{\tilde{O}_1, \tilde{O}_2\}$ is a pure pair with latent parent $\mathrm{Pa}_{\mathbf{L}}^{\mathcal{G}}(\mathcal{S})$ where $\tilde{O}_1 = O_1$ and $\tilde{O}_2 = O_2 + \frac{1}{\alpha} O_1$.*

*Proof.* Based on Lemma 3, $\mathcal{S} = \{O_1, O_2\}$ is a pseudo-pure pair. Suppose $O_2 \in \mathrm{Pa}^{\mathcal{G}}(O_1)$, then $O_1$ contains $\epsilon_{O_1}$ while $\forall O \notin \mathbf{O} \backslash \{O_1\}$, $O$ does not contain $\epsilon_{O_1}$, so $\forall \alpha, \beta$, $O_1 + \alpha O_2 + \beta O_3$ contains $\epsilon_{O_1}$. As $\epsilon_{O_1}$ is non-Gaussian based on Assumption 3(a), $O_1 + \alpha O_2 + \beta O_3 \not\perp\!\!\!\perp O_1$, which leads to contradiction. Therefore, we can conclude $O_1 \in \mathrm{Pa}^{\mathcal{G}}(O_2)$. Let $\mathrm{Pa}_{\mathbf{L}}^{\mathcal{G}}(\mathcal{S}) = \{L\}$, then $O_1, O_2$ can be represented as

$$O_1 = c_{11} L + \epsilon_{O_1}, \tag{25}$$

$$O_2 = c_{12} L + d_{12} O_1 + \epsilon_{O_2} = (c_{11} d_{12} + c_{12}) L + d_{12} \epsilon_{O_1} + \epsilon_{O_2}. \tag{26}$$

We can easily obtain $\alpha = -\frac{1}{d_{12}}$, because if $\alpha \neq -\frac{1}{d_{12}}$, both $O_1$ and $O_1 + \alpha O_2 + \beta O_3$ contain non-Gaussian $\epsilon_{O_1}$, i.e., $O_1 + \alpha O_2 + \beta O_3 \not\perp\!\!\!\perp O_1$. Therefore, we can represent $\tilde{O}_1, \tilde{O}_2$ as

$$\tilde{O}_1 = c_{11} L + \epsilon_{O_1}, \quad \tilde{O}_2 = c_{12} L + \epsilon_{O_2}. \tag{27}$$

Clearly, $\{\tilde{O}_1, \tilde{O}_2\}$ is a pure pair with latent parent $L$. $\qquad\qquad\square$

### B.10 PROOF OF PROPOSITION 2

**Proposition 2.** *Let $\{\mathcal{S}_1, \mathcal{S}_2\} \subset \mathbb{S}$ where $\mathcal{S}_1 = \{O_1, O_2\}$ and $\mathcal{S}_2 = \{O_3, O_4\}$. Then*

1. *Suppose $\mathcal{S}_1$ and $\mathcal{S}_2$ are two pure pairs. $\mathrm{Pa}_{\mathbf{L}}^{\mathcal{G}}(\mathcal{S}_1) = \mathrm{Pa}_{\mathbf{L}}^{\mathcal{G}}(\mathcal{S}_2)$ if and only if (1) $\mathcal{S}_1 \cap \mathcal{S}_2 \neq \emptyset$, or (2) $\exists \mathcal{S}_3 \in \mathbb{S}$ s.t. $\mathcal{S}_1 \cap \mathcal{S}_3 \neq \emptyset$ and $\mathcal{S}_2 \cap \mathcal{S}_3 \neq \emptyset$.*

2. *Suppose $\mathcal{S}_1$ is a pure pair and $\mathcal{S}_2$ is a pseudo-pure pair. Then $\mathrm{Pa}_{\mathbf{L}}^{\mathcal{G}}(\mathcal{S}_1) = \mathrm{Pa}_{\mathbf{L}}^{\mathcal{G}}(\mathcal{S}_2)$ if and only if $(\{O_2, \tilde{O}_3\}, \{O_1, \tilde{O}_4\})$ satisfies the tetrad constraint.*

3. *Suppose $\mathcal{S}_1$ and $\mathcal{S}_2$ are two pseudo-pure pairs. Then $\mathrm{Pa}_{\mathbf{L}}^{\mathcal{G}}(\mathcal{S}_1) = \mathrm{Pa}_{\mathbf{L}}^{\mathcal{G}}(\mathcal{S}_2)$ if and only if $(\{\tilde{O}_2, \tilde{O}_3\}, \{\tilde{O}_1, \tilde{O}_4\})$ satisfies the tetrad constraint.*

*Proof.* (i) "If".

1. Suppose $\mathcal{S}_1$ and $\mathcal{S}_2$ are two pure pairs. The proof is the same as that of the first sub-proposition of Proposition 1.

2. Suppose $\mathcal{S}_1$ is a pure pair and $\mathcal{S}_2$ is a pseudo-pure pair. Clearly, $\mathcal{S}_1 \cap \mathcal{S}_2 = \emptyset$. Based on Corollary 2, $\{\tilde{O}_3, \tilde{O}_4\}$ is a pure pair with latent parent $\mathrm{Pa}_{\mathbf{L}}^{\mathcal{G}}(\mathcal{S}_2)$. This part can be proved by contradiction. Suppose $\mathrm{Pa}_{\mathbf{L}}^{\mathcal{G}}(\mathcal{S}_1) \neq \mathrm{Pa}_{\mathbf{L}}^{\mathcal{G}}(\mathcal{S}_2)$, let $\mathrm{Pa}_{\mathbf{L}}^{\mathcal{G}}(\mathcal{S}_1) = \{L_1\}$ and $\mathrm{Pa}_{\mathbf{L}}^{\mathcal{G}}(\mathcal{S}_2) = \{L_2\}$. Then there are two non-intersecting treks $O_1 \leftarrow L_1 \rightarrow O_2$ and $\tilde{O}_3 \leftarrow L_2 \rightarrow \tilde{O}_4$, so there is no choke point between $\{O_2, \tilde{O}_3\}$ and $\{O_1, \tilde{O}_4\}$. Therefore, $(\{O_2, \tilde{O}_3\}, \{O_1, \tilde{O}_4\})$ **does not** satisfy the tetrad constraint, which leads to contradiction.

3. Suppose $\mathcal{S}_1$ and $\mathcal{S}_2$ are two pseudo-pure pairs. The remaining proof is similar to that of the second sub-proposition above.

(ii) "Only if".

1. Suppose $\mathcal{S}_1$ and $\mathcal{S}_2$ are two pure pairs. The proof is the same as that of the first sub-proposition of Proposition 1.

2. Suppose $\mathcal{S}_1$ is a pure pair and $\mathcal{S}_2$ is a pseudo-pure pair. Clearly, $\mathcal{S}_1 \cap \mathcal{S}_2 = \emptyset$. Based on Corollary 2, $\{\tilde{O}_3, \tilde{O}_4\}$ is a pure pair with latent parent $\mathrm{Pa}_{\mathbf{L}}^{\mathcal{G}}(\mathcal{S}_2)$. Suppose $\mathrm{Pa}_{\mathbf{L}}^{\mathcal{G}}(\mathcal{S}_1) = \mathrm{Pa}_{\mathbf{L}}^{\mathcal{G}}(\mathcal{S}_2)$, let $\mathrm{Pa}_{\mathbf{L}}^{\mathcal{G}}(\mathcal{S}_1) = \{L\}$. Since $O_1, O_2, \tilde{O}_3, \tilde{O}_4$ are all pure children of $L$, $L$ is a choke point between $\{O_2, \tilde{O}_3\}$ and $\{O_1, \tilde{O}_4\}$. Therefore, $(\{O_2, \tilde{O}_3\}, \{O_1, \tilde{O}_4\})$ satisfies the tetrad constraint.

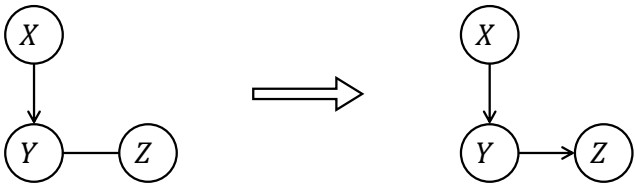

Figure 8: Illustration of Meek's rule 1

3. Suppose $\mathcal{S}_1$ and $\mathcal{S}_2$ are two pseudo-pure pairs. The remaining proof is similar to that of the second sub-proposition above.

$\square$

### B.11 PROOF OF THEOREM 3

**Theorem 3.** *Suppose the underlying linear latent variable model satisfies Assumption 1 and 3. Then latent variables can be fully identified.*

*Proof.* Theorem 1 ensures no latent omission under Assumption 1 while Proposition 2 ensures no latent commission under Assumption 1 and 3. Therefore, latent variables can be fully identified. $\square$

### B.12 PROOF OF PROPOSITION 3

**Proposition 3.** *No variable in $\mathbf{O}^D$ is a parent of any variable in $\mathbf{L} \cup \mathbf{O}^U$.*

*Proof.* We prove this proposition in Case I (where Assumption 1 and 2 hold) and Case II (where Assumption 1 and 3 hold) respectively.

1. In Case I, $O_1 \in \mathbf{O}^D$ if and only if (1) $\exists O_2 \in \mathbf{O}$ s.t. $\{O_1, O_2\} \in \mathbb{S}$, or (2) $\exists \mathcal{S} \in \mathbb{S}$ s.t. $O_1 = \mathrm{Ref}(\mathcal{S})$. Based on Corollary 1, the latter means that $O_1$ is a pure child of some latent variable. Therefore, we can easily reach the conclusion that no variable in $\mathbf{O}^D$ is a parent of any variable in $\mathbf{L} \cup \mathbf{O}^U$.

2. In Case II, $O_1 \in \mathbf{O}^D$ if and only if $\exists O_2 \in \mathbf{O}$ s.t. $\{O_1, O_2\} \in \mathbb{S}$, so no variable in $\mathbf{O}^D$ is a parent of any variable in $\mathbf{L} \cup \mathbf{O}^U$.

$\square$

### B.13 PROOF OF THEOREM 4

**Theorem 4.** *Suppose the underlying linear latent variable model satisfies Assumption 1 and 2 or Assumption 1 and 3, in the limit of infinite data, $\hat{\mathcal{G}}$ satisfies that (1) $\hat{\mathcal{G}}$ has the same skeleton and v-structures as $\mathcal{G}$; (2) $\forall \{O_i, O_j\} \subset \mathbf{O}$ s.t. $O_i \in \mathrm{Pa}^{\mathcal{G}}(O_j)$ and $\mathrm{Pa}_{\mathbf{L}}^{\mathcal{G}}(O_i) \neq \mathrm{Pa}_{\mathbf{L}}^{\mathcal{G}}(O_j)$, $O_i \in \mathrm{Pa}^{\hat{\mathcal{G}}}(O_j)$.*

*Proof.* Based on Theorem 2 and Theorem 3, latent variables can be fully identified. After pre-processing, each latent variable has multiple indicators of which each can be represented as its linear function plus an independent noise.

(1) This is derived by the soundness of PC algorithm.

(2) Without loss of generality, suppose $O_i \in \mathrm{Pa}^{\mathcal{G}}(O_j)$, there are two possible cases w.r.t. $\mathrm{Pa}_{\mathbf{L}}^{\mathcal{G}}(O_j)$.

1. Suppose $\exists L \in \mathrm{Pa}_{\mathbf{L}}^{\mathcal{G}}(O_j)$ s.t. $L \notin \mathrm{Pa}_{\mathbf{L}}^{\mathcal{G}}(O_i)$. Then there exists a v-structure $O_i \to O_j \leftarrow L$ in $\mathcal{G}$, which can be discovered by line 2 of Algorithm 4.

2. Suppose $\forall L \in \mathrm{Pa}_{\mathbf{L}}^{\mathcal{G}}(O_j), L \in \mathrm{Pa}_{\mathbf{L}}^{\mathcal{G}}(O_i)$. Since $\mathrm{Pa}_{\mathbf{L}}^{\mathcal{G}}(O_i) \neq \mathrm{Pa}_{\mathbf{L}}^{\mathcal{G}}(O_j)$, $\exists L' \in \mathrm{Pa}_{\mathbf{L}}^{\mathcal{G}}(O_i)$ s.t. $L' \notin \mathrm{Pa}_{\mathbf{L}}^{\mathcal{G}}(O_j)$. After line 3 of Algorithm 4, there is $L' \to O_i - O_j$ and $L'$ is not adjacent to $O_j$ in $\hat{\mathcal{G}}$. Based on Meek's rule 1 shown as Figure 8, $O_i - O_j$ can be oriented as $O_i \to O_j$.

$\square$

## C MORE THEORETICAL RESULTS

### C.1 NON-LINEARITY

All theoretical results in Section 3.1 and 3.2 are derived by the Tetrad Representation Theorem (Spirtes et al., 2000), which is a special form of the Trek Separation Theorem (Sullivant et al., 2010). Furthermore, Spirtes (2013) has extended the Trek Separation Theorem to partially nonlinear cases. An exact formulation of the Extended Trek Separation Theorem entails many concepts not previously introduced, here we only need to know

1. If there is a choke point $C$ between $\{I_1, I_2\}$ and $\{J_1, J_2\}$, $C$ is on the $\{I_1, I_2\}$ side, and for each directed path $\pi$ from $C$ to $\{I_1, I_2\}$, any vertex $V$ on $\pi$ is a linear function of its parents along $\pi$ plus an arbitrary function of the parents not along $\pi$, then $(\{I_1, I_2\}, \{J_1, J_2\})$ satisfies the tetrad constraint.

2. If there is no choke point between $\{I_1, I_2\}$ and $\{J_1, J_2\}$, under faithfulness assumption, $(\{I_1, I_2\}, \{J_1, J_2\})$ does not satisfy the tetrad constraint.

Based on these two propositions, we can conclude that all theoretical results in Section 3.1 and 3.2 are still valid as long as all causal relations involving generalized pure children (see Definition 9) are linear. More specifically, any causal relation between $V_1$ and $V_2$ can be nonlinear as long as neither $V_1$ nor $V_2$ is a generalized pure child of some latent variable. Any tetrad constraint in this case holds if and only if it holds in the linear case. Furthermore, if only causal relations between observed variables that are not pseudo-pure children (see Definition 8) of some latent variable are nonlinear while others are all linear, all theoretical results in Section 3.3 also hold, because our proofs do not rely on linearity of these causal relations.

### C.2 WEAKENING ASSUMPTION 1(B)

Assumption 1(b) requires that $\forall L \in \mathbf{L}, \mathrm{Nei}_{\mathbf{L}}^{\mathcal{G}}(L) \neq \emptyset$. It is used only in the proof of Theorem 1 and Lemma 2. In fact, even if $\exists L \in \mathbf{L}$ s.t. $\mathrm{Nei}_{\mathbf{L}}^{\mathcal{G}}(L) = \emptyset$, Theorem 1 still holds if for every such $L$, $|\mathrm{Ch}_{\mathbf{O}}^{\mathcal{G}}(L)| \geq 4$; and Lemma 2 still holds if for every such $L$, $|\mathrm{Ch}_{\mathbf{O}}^{\mathcal{G}}(L)| \geq 5$. Taking Lemma 2 as an example, we prove that it still holds if $\forall L \in \mathbf{L}$ s.t. $\mathrm{Nei}_{\mathbf{L}}^{\mathcal{G}}(L) = \emptyset, |\mathrm{Ch}_{\mathbf{O}}^{\mathcal{G}}(L)| \geq 5$.

*Proof.* In the proof of Lemma 2 given in Appendix B.3, Assumption 1(b) is only used to prove that $\forall O \in \mathbf{O}^{\mathrm{C}} \backslash \{O_1, O_2\}, (\{O_1, O\}, \mathbf{O} \backslash \{O_1, O_2, O\})$ **does not** satisfy the tetrad constraint if $\mathcal{S} = \{O_1, O_2\}$ is a pure pair and $\mathbb{1}_{\mathrm{pure}}(\mathcal{S}) = -1$. We only need to prove this part. Let $\mathrm{Pa}_{\mathbf{L}}^{\mathcal{G}}(\mathcal{S}) = \{L_1\}$, $L_1$ has no pure child except $O_1$ and $O_2$ since $\mathbb{1}_{\mathrm{pure}}(\mathcal{S}) = -1$. Given an $O_3 \in \mathbf{O}^{\mathrm{C}} \backslash \{O_1, O_2\}$, there are three possible cases w.r.t. $\mathrm{Pa}_{\mathbf{L}}^{\mathcal{G}}(O_3)$.

1. Suppose $\mathrm{Pa}_{\mathbf{L}}^{\mathcal{G}}(O_3) = \emptyset$. Based on Lemma 5, $O_3$ has an observed children $O_4 \in \mathbf{O} \backslash \{O_1, O_2\}$. If $\mathrm{Nei}_{\mathbf{L}}^{\mathcal{G}}(L_1) \neq \emptyset$, the proof is the same as the original one; otherwise, there is $|\mathrm{Ch}_{\mathbf{O}}^{\mathcal{G}}(L_1)| \geq 5$ according to our new assumption, so $\exists O_5 \in \mathrm{Ch}_{\mathbf{O}}^{\mathcal{G}}(L_1) \backslash \{O_1, O_2, O_3, O_4\}$. Clearly, there are two non-intersecting treks $O_3 - O_4$ and $O_1 \leftarrow L_1 \to O_5$. Therefore, there is no choke point between $\{O_1, O_3\}$ and $\{O_4, O_5\}$.

2. Suppose $\mathrm{Pa}_{\mathbf{L}}^{\mathcal{G}}(O_3) = \{L_1\}$. Since $O_3$ is not a pure child of $L_1$, $\exists O_4 \in \mathrm{Nei}_{\mathbf{O}}^{\mathcal{G}}(O_3)$ where $O_4 \notin \{O_1, O_2\}$. If $\mathrm{Nei}_{\mathbf{L}}^{\mathcal{G}}(L_1) \neq \emptyset$, the proof is the same as the original one; otherwise, there is $|\mathrm{Ch}_{\mathbf{O}}^{\mathcal{G}}(L_1)| \geq 5$ according to our new assumption, so $\exists O_5 \in \mathrm{Ch}_{\mathbf{O}}^{\mathcal{G}}(L_1) \backslash \{O_1, O_2, O_3, O_4\}$.

Clearly, there are two non-intersecting treks $O_3 - O_4$ and $O_1 \leftarrow L_1 \rightarrow O_5$. Therefore, there is no choke point between $\{O_1, O_3\}$ and $\{O_4, O_5\}$.

3. Suppose $\text{Pa}_{\mathbf{L}}^{\mathcal{G}}(O_3) \neq \emptyset$ and $\text{Pa}_{\mathbf{L}}^{\mathcal{G}}(O_3) \neq \{L_1\}$, that is, $\exists L_2 \in \text{Pa}_{\mathbf{L}}^{\mathcal{G}}(O_3) \backslash \{L_1\}$. Then $L_2$ has a generalized pure child $O_4 \in \mathbf{O} \backslash \{O_1, O_2, O_3\}$ according to Assumption 1(a). If $\text{Nei}_{\mathbf{L}}^{\mathcal{G}}(L_1) \neq \emptyset$, the proof is the same as the original one; otherwise, there is $|\text{Ch}_{\mathbf{O}}^{\mathcal{G}}(L_1)| \geq 5$ according to our new assumption, so $\exists O_5 \in \text{Ch}_{\mathbf{O}}^{\mathcal{G}}(L_1) \backslash \{O_1, O_2, O_3, O_4\}$. Clearly, there are two non-intersecting treks $O_3 \leftarrow L_2 \rightarrow O_4$ and $O_1 \leftarrow L_1 \rightarrow O_5$. Therefore, there is no choke point between $\{O_1, O_3\}$ and $\{O_4, O_5\}$.

Based on Theorem 5, we reach the conclusion that $\forall O \in \mathbf{O}^C \backslash \{O_1, O_2\}$, $(\{O_1, O\}, \mathbf{O} \backslash \{O_1, O_2, O\})$ **does not** satisfy the tetrad constraint. □

## D   DETAILS OF ALGORITHMS

The detailed versions of Algorithm 1, 2, and 3 are shown as Algorithm 5, 6, and 7. Besides, we also provide two illustrative examples to show how each step proceeds.

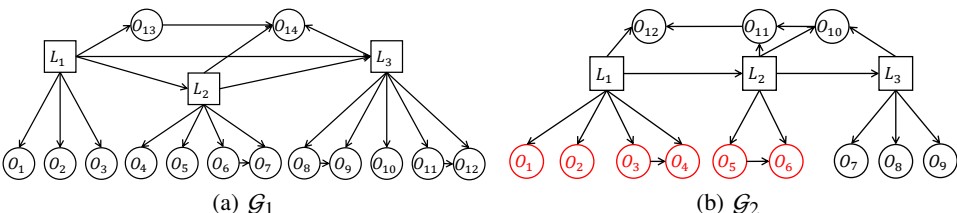

(a) $\mathcal{G}_1$                              (b) $\mathcal{G}_2$

Taking the causal model with structure $\mathcal{G}_1$ as an example of Case I, we show the procedures of recovering the whole causal graph as follows.

1. Line 1 of Algorithm 1: find all candidate variables. Here $\mathbf{O}^C = \mathbf{O}$.

2. Line 2 of Algorithm 1: find all generalized pure pairs. Here $\mathbb{S} = \{\{O_1, O_2\}, \{O_1, O_3\}, \{O_2, O_3\}, \{O_4, O_5\}, \{O_6, O_7\}, \{O_8, O_9\}, \{O_{11}, O_{12}\}\}$.

3. Line 3 of Algorithm 1: identify as many pure pairs as possible. Since $\{O_1, O_2\} \cap \{O_1, O_3\} \neq \emptyset$ and $\{O_1, O_2\} \cap \{O_2, O_3\} \neq \emptyset$, $\mathbb{1}_{\text{pure}}(\{O_1, O_2\}) = \mathbb{1}_{\text{pure}}(\{O_1, O_3\}) = \mathbb{1}_{\text{pure}}(\{O_2, O_3\}) = 1$, for any other generalized pure pair $\mathbb{1}_{\text{pure}}(\cdot)$ is -1.

4. Line 1 of Algorithm 2: discriminate pure pairs against pseudo-pure ones. Since $(\{O_6, O_4\}, \mathbf{O} \backslash \{O_4, O_6, O_7\})$ satisfies the tetrad constraint, $\mathbb{1}_{\text{pure}}(\{O_6, O_7\}) = 0$ and $\text{Ref}(\{O_6, O_7\}) = O_4$. Similarly, we have $\mathbb{1}_{\text{pure}}(\{O_8, O_9\}) = 0, \mathbb{1}_{\text{pure}}(\{O_{11}, O_{12}\}) = 0$ and $\text{Ref}(\{O_8, O_9\}) = O_{10}, \text{Ref}(\{O_{11}, O_{12}\}) = O_{10}$. Besides, $\mathbb{1}_{\text{pure}}(\{O_4, O_5\}) = 1$

5. Line 2 of Algorithm 2: check whether any two generalized pure pairs share a common latent parent. Clearly, $\text{Pa}_{\mathbf{L}}^{\mathcal{G}}(\{O_1, O_2\}) = \text{Pa}_{\mathbf{L}}^{\mathcal{G}}(\{O_1, O_3\}) = \text{Pa}_{\mathbf{L}}^{\mathcal{G}}(\{O_2, O_3\}) = \{L_1\}$, $\text{Pa}_{\mathbf{L}}^{\mathcal{G}}(\{O_4, O_5\}) = \text{Pa}_{\mathbf{L}}^{\mathcal{G}}(O_4) = \text{Pa}_{\mathbf{L}}^{\mathcal{G}}(\{O_6, O_7\}) = \{L_2\}$, and $\text{Pa}_{\mathbf{L}}^{\mathcal{G}}(\{O_8, O_9\}) = \text{Pa}_{\mathbf{L}}^{\mathcal{G}}(O_{10}) = \text{Pa}_{\mathbf{L}}^{\mathcal{G}}(\{O_{11}, O_{12}\}) = \{L_3\}$.

6. Pre-processing in Section 4.1: $\mathbf{L} = \{L_1, L_2, L_3\}$, $\mathbf{O}^D = \{O_i\}_{i=1}^{12}$, $\mathbf{O}^U = \{O_{13}, O_{14}\}$. The measured indicators of $L_1, L_2$ and $L_3$ can be respectively $\{O_1, O_2\}, \{O_4, O_5\}$ and $\{O_8, O_{10}\}$. Furthermore, we also create two auxiliary measured indicators for each variable in $\mathbf{O}^U$.

7. Run Algorithm 4 to reveal causal relations between any two variables.

Taking the causal model with structure $\mathcal{G}_2$ as an example of Case II, we show the procedures of recovering the whole causal graph as follows.

1. Line 1 of Algorithm 1: find all candidate variables $\mathbf{O}^C$. Here $\mathbf{O}^C = \mathbf{O}$.

2. Line 2 of Algorithm 1: find all generalized pure pairs $\mathbb{S}$. Here $\mathbb{S} = \{\{O_1, O_2\}, \{O_3, O_4\}, \{O_5, O_6\}, \{O_7, O_8\}, \{O_7, O_9\}, \{O_8, O_9\}\}$.

3. Line 3 of Algorithm 1: identify as many pure pairs as possible. Since $\{O_7, O_8\} \cap \{O_7, O_9\} \neq \emptyset$ and $\{O_7, O_8\} \cap \{O_8, O_9\} \neq \emptyset$, $\mathbb{1}_{\text{pure}}(\{O_7, O_8\}) = \mathbb{1}_{\text{pure}}(\{O_7, O_9\}) = \mathbb{1}_{\text{pure}}(\{O_8, O_9\}) = 1$, for any other generalized pure pair $\mathbb{1}_{\text{pure}}(\cdot)$ is -1.

4. Line 1 of Algorithm 3: discriminate pure pairs against pseudo-pure ones and convert each pseudo-pure ones into a pure one. Since $(O_1, O_2)$ can make $O_3 + \alpha O_4 + \beta O_1 \perp\!\!\!\perp O_3$ hold and $(O_7, O_1)$ can make $O_5 + \alpha O_6 + \beta O_7 \perp\!\!\!\perp O_5$ hold, we have $\mathbb{1}_{\text{pure}}(\{O_3, O_4\}) = 0$ and $\mathbb{1}_{\text{pure}}(\{O_5, O_6\}) = 0$. Besides, we also have $\mathbb{1}_{\text{pure}}(\{O_1, O_2\}) = 1$. Then we convert $\{O_3, O_4\}$ into $\{\tilde{O}_3, \tilde{O}_4\}$ and $\{O_5, O_6\}$ into $\{\tilde{O}_5, \tilde{O}_6\}$

5. Line 2 of Algorithm 3: check whether two generalized pure pairs share a common latent parent. We have $\text{Pa}_{\mathbf{L}}^{\mathcal{G}}(\{O_1, O_2\}) = \text{Pa}_{\mathbf{L}}^{\mathcal{G}}(\{O_3, O_4\}) = \{L_1\}$ since $(\{O_1, \tilde{O}_3\}, \{O_2, \tilde{O}_4\})$ satisfies the tetrad constraint, $\text{Pa}_{\mathbf{L}}^{\mathcal{G}}(\{O_5, O_6\}) = \{L_2\}$, and $\text{Pa}_{\mathbf{L}}^{\mathcal{G}}(\{O_7, O_8\}) = \text{Pa}_{\mathbf{L}}^{\mathcal{G}}(\{O_7, O_9\}) = \text{Pa}_{\mathbf{L}}^{\mathcal{G}}(\{O_8, O_9\}) = \{L_3\}$.

6. Pre-processing in Section 4.1: $\mathbf{L} = \{L_1, L_2, L_3\}$, $\mathbf{O}^{\text{D}} = \{O_i\}_{i=1}^{9}$, $\mathbf{O}^{\text{U}} = \{O_{10}, O_{11}, O_{12}\}$. The measured indicators of $L_1, L_2$ and $L_3$ can be respectively $\{O_1, O_2\}$, $\{\tilde{O}_5, \tilde{O}_6\}$ and $\{O_7, O_8\}$. Furthermore, we also create two auxiliary measured indicators for each variable in $\mathbf{O}^{\text{U}}$.

7. Run Algorithm 4 to reveal causal relations between any two variables.

---

**Algorithm 5:** Partially identifying latent variables under Assumption 1 (a detailed version).

---

**Input:** Observed variable $\mathbf{O}$.
**Output:** Candidate variables $\mathbf{O}^{\mathrm{C}}$, generalized pure pairs $\mathbb{S}$, purity indicator function $\mathbb{1}_{\mathrm{pure}}(\cdot)$.

1   // Find all candidate variables.
2   $\mathbf{O}^{\mathrm{C}} := \emptyset$;
3   **for** $O_1 \in \mathbf{O}$ **do**
4      flag := 1.
5      **for** $\{O_2, O_3\} \subset \mathbf{O}\backslash\{O_1\}$ **do**
6         **if** $\forall O_4 \in \mathbf{O}\backslash\{O_1, O_2, O_3\}$ *s.t.* $O_1 \perp\!\!\!\perp O_4$ *given some subset of* $\{O_2, O_3\}$ **then**
7            flag := 0;
8            **break**
9         **end**
10      **end**
11      **if** *flag = 1* **then**
12         $\mathbf{O}^{\mathrm{C}} := \mathbf{O}^{\mathrm{C}} \cup \{O_1\}$;
13      **end**
14   **end**
15   // Find all generalized pure pairs.
16   $\mathbb{S} = \emptyset$;
17   **for** $\{O_1, O_2\} \subset \mathbf{O}^{\mathrm{C}}$ **do**
18      **if** $\forall \{O_3, O_4\} \subset \mathbf{O}\backslash\{O_1, O_2\}, \mathrm{Cov}(O_1, O_4)\mathrm{Cov}(O_2, O_3) = \mathrm{Cov}(O_1, O_3)\mathrm{Cov}(O_2, O_4)$ **then**
19         $\mathbb{S} := \mathbb{S} \cup \{\{O_1, O_2\}\}$
20      **end**
21   **end**
22   // Identify as many pure pairs as possible;
23   **for** $\mathcal{S} \in \mathbb{S}$ **do**
24      $\mathbb{1}_{\mathrm{pure}}(\mathcal{S}) := -1$;
25   **end**
26   **for** $\mathcal{S} \in \mathbb{S}$ **do**
27      **if** $\exists \mathcal{S}' \in \mathbb{S}\backslash\{\mathcal{S}\}$ *s.t.* $\mathcal{S} \cap \mathcal{S}' \neq \emptyset$ **then**
28         $\mathbb{1}_{\mathrm{pure}}(\mathcal{S}) := 1$
29      **end**
30   **end**

---

---

**Algorithm 6:** Fully identifying latent variables in Case I (a detailed version).

---

**Input:** Observed variables $\mathbf{O}$, candidate variables $\mathbf{O}^C$, generalized pure pairs $\mathbb{S}$, purity indicator function $\mathbb{1}_{\text{pure}}(\cdot)$

**Output:** Updated purity indicator function $\mathbb{1}_{\text{pure}}(\cdot)$, sibling indicator function $\mathbb{1}_{\text{sib}}(\cdot, \cdot)$.

**1** // Discriminate pure pairs against pseudo-pure pairs.

**2 for** $\mathcal{S} = \{O_1, O_2\} \subset \mathbb{S}$ s.t. $\mathbb{1}_{\text{pure}}(\mathcal{S}) = -1$ **do**

**3**     **for** $O_3 \in \mathbf{O}^C \backslash \{O_1, O_2\}$ **do**

**4**         **if** $\forall \{O_4, O_5\} \subset \mathbf{O} \backslash \{O_1, O_2, O_3\}, \text{Cov}(O_1, O_5)\text{Cov}(O_3, O_4) = \text{Cov}(O_1, O_4)\text{Cov}(O_3, O_5)$
          **then**

**5**             $\mathbb{1}_{\text{pure}}(\mathcal{S}) := 0$;

**6**             $\text{Ref}(\mathcal{S}) := O_3$;

**7**         **else**

**8**             $\mathbb{1}_{\text{pure}}(\mathcal{S}) := 1$;

**9**         **end**

**10**     **end**

**11 end**

**12** // Check whether two generalized pure pairs share a common latent parent.

**13 for** $\{\mathcal{S}_1, \mathcal{S}_2\} \subset \mathbb{S}$ **do**

**14**     $\mathbb{1}_{\text{sib}}(\mathcal{S}_1, \mathcal{S}_2) := 0$;

**15 end**

**16 for** $\{\mathcal{S}_1, \mathcal{S}_2\} \subset \mathbb{S}$ **do**

**17**     **if** $\mathbb{1}_{\text{pure}}(\mathcal{S}_1) = 1$ and $\mathbb{1}_{\text{pure}}(\mathcal{S}_2) = 1$ **then**

**18**         **if** $\mathcal{S}_1 \cap \mathcal{S}_2 \neq \emptyset$ or $\exists \mathcal{S}_3 \in \mathbb{S} \backslash \{\mathcal{S}_1, \mathcal{S}_2\}$ s.t. $\mathcal{S}_1 \cap \mathcal{S}_3 \neq \emptyset$ *and* $\mathcal{S}_2 \cap \mathcal{S}_3 \neq \emptyset$ **then**

**19**             $\mathbb{1}_{\text{sib}}(\mathcal{S}_1, \mathcal{S}_2) := 1$;

**20**         **end**

**21**     **else if** $\mathbb{1}_{\text{pure}}(\mathcal{S}_1) = 0$ and $\mathbb{1}_{\text{pure}}(\mathcal{S}_2) = 1$ **then**

**22**         **if** $\text{Ref}(\mathcal{S}_1) \in \mathcal{S}_2$ or $\exists \mathcal{S}_3 \in \mathbb{S} \backslash \{\mathcal{S}_1, \mathcal{S}_2\}$ s.t. $\text{Ref}(\mathcal{S}_1) \in \mathcal{S}_3$ *and* $\mathcal{S}_2 \cap \mathcal{S}_3 \neq \emptyset$ **then**

**23**             $\mathbb{1}_{\text{sib}}(\mathcal{S}_1, \mathcal{S}_2) := 1$;

**24**         **end**

**25**     **else if** $\mathbb{1}_{\text{pure}}(\mathcal{S}_1) = 1$ and $\mathbb{1}_{\text{pure}}(\mathcal{S}_2) = 0$ **then**

**26**         **if** $\text{Ref}(\mathcal{S}_2) \in \mathcal{S}_1$ or $\exists \mathcal{S}_3 \in \mathbb{S} \backslash \{\mathcal{S}_1, \mathcal{S}_2\}$ s.t. $\text{Ref}(\mathcal{S}_2) \in \mathcal{S}_3$ *and* $\mathcal{S}_1 \cap \mathcal{S}_3 \neq \emptyset$ **then**

**27**             $\mathbb{1}_{\text{sib}}(\mathcal{S}_1, \mathcal{S}_2) := 1$;

**28**         **end**

**29**     **else**

**30**         **if** $\text{Ref}(\mathcal{S}_1) \in \text{Ref}(\mathcal{S}_2)$ or $\exists \mathcal{S}_3 \in \mathbb{S} \backslash \{\mathcal{S}_1, \mathcal{S}_2\}$ s.t. $\text{Ref}(\mathcal{S}_1) \in \mathcal{S}_3$ *and* $\text{Ref}(\mathcal{S}_2) \in \mathcal{S}_3$ **then**

**31**             $\mathbb{1}_{\text{sib}}(\mathcal{S}_1, \mathcal{S}_2) := 1$;

**32**         **end**

**33**     **end**

**34 end**

---

---

**Algorithm 7:** Fully identifying latent variables in Case II (a detailed version).

---

**Input:** Observed variables $\mathbf{O}$, generalized pure pairs $\mathbb{S}$, purity indicator function $\mathbb{1}_{\text{pure}}(\cdot)$
**Output:** Updated purity indicator function $\mathbb{1}_{\text{pure}}(\cdot)$, sibling indicator function $\mathbb{1}_{\text{sib}}(\cdot, \cdot)$.

1  // Discriminate pure pairs against pseudo-pure pairs.
2  **for** $\mathcal{S} = \{O_1, O_2\} \subset \mathbb{S}$ s.t. $\mathbb{1}_{\text{pure}}(\mathcal{S}) = -1$ **do**
3      **for** $(O_3, O_4) \subset \mathbf{O} \backslash \{O_1, O_2\}$ **do**
4          **if** $O_1 + \alpha O_2 + \beta O_3 \perp\!\!\!\perp O_1$ *where* $\alpha, \beta$ *satisfy Equation (2) and (3)* **then**
5              $\mathbb{1}_{\text{pure}}(\mathcal{S}) =: 0, \tilde{O}_1 := O_1, \tilde{O}_2 := O_2 + \frac{1}{\alpha}O_1$;
6          **else if** $O_2 + \alpha O_1 + \beta O_3 \perp\!\!\!\perp O_2$ *where* $\alpha, \beta$ *satisfy Equation (4) and (5)* **then**
7              $\mathbb{1}_{\text{pure}}(\mathcal{S}) =: 0, \tilde{O}_1 := O_1 + \frac{1}{\alpha}O_2, \tilde{O}_2 := O_2$;
8          **else**
9              $\mathbb{1}_{\text{pure}}(\mathcal{S}) =: 1$;
10         **end**
11     **end**
12 **end**
13 // Check whether two generalized pure pairs share a common latent parent.
14 **for** $\{\mathcal{S}_1, \mathcal{S}_2\} \subset \mathbb{S}$ **do**
15     $\mathbb{1}_{\text{sib}}(\mathcal{S}_1, \mathcal{S}_2) := 0$;
16 **end**
17 **for** $\{\mathcal{S}_1, \mathcal{S}_2\} \subset \mathbb{S}$ **do**
18     **if** $\mathbb{1}_{\text{pure}}(\mathcal{S}_1) = 1$ and $\mathbb{1}_{\text{pure}}(\mathcal{S}_2) = 1$ **then**
19         **if** $\mathcal{S}_1 \cap \mathcal{S}_2 \neq \emptyset$ or $\exists \mathcal{S}_3 \in \mathbb{S} \backslash \{\mathcal{S}_1, \mathcal{S}_2\}$ s.t. $\mathcal{S}_1 \cap \mathcal{S}_3 \neq \emptyset$ *and* $\mathcal{S}_2 \cap \mathcal{S}_3 \neq \emptyset$ **then**
20             $\mathbb{1}_{\text{sib}}(\mathcal{S}_1, \mathcal{S}_2) := 1$;
21         **end**
22     **else if** $\mathbb{1}_{\text{pure}}(\mathcal{S}_1) = 0$ and $\mathbb{1}_{\text{pure}}(\mathcal{S}_2) = 1$ **then**
23         **if** $\text{Cov}(\tilde{O}_1, O_4)\text{Cov}(\tilde{O}_2, O_3) = \text{Cov}(\tilde{O}_1, O_3)\text{Cov}(\tilde{O}_2, O_4)$ **then**
24             $\mathbb{1}_{\text{sib}}(\mathcal{S}_1, \mathcal{S}_2) := 1$;
25         **end**
26     **else if** $\mathbb{1}_{\text{pure}}(\mathcal{S}_1) = 1$ and $\mathbb{1}_{\text{pure}}(\mathcal{S}_2) = 0$ **then**
27         **if** $\text{Cov}(O_1, \tilde{O}_4)\text{Cov}(O_2, \tilde{O}_3) = \text{Cov}(O_1, \tilde{O}_3)\text{Cov}(O_2, \tilde{O}_4)$ **then**
28             $\mathbb{1}_{\text{sib}}(\mathcal{S}_1, \mathcal{S}_2) := 1$;
29         **end**
30     **else**
31         **if** $\text{Cov}(\tilde{O}_1, \tilde{O}_4)\text{Cov}(\tilde{O}_2, \tilde{O}_3) = \text{Cov}(\tilde{O}_1, \tilde{O}_3)\text{Cov}(\tilde{O}_2, \tilde{O}_4)$ **then**
32             $\mathbb{1}_{\text{sib}}(\mathcal{S}_1, \mathcal{S}_2) := 1$;
33         **end**
34     **end**
35 **end**

---

