# OpenReview forum: "Causal Structure Recovery with Latent Variables under Milder Distributional and Graphical Assumptions"
_ICLR.cc/2024/Conference — ICLR 2024 poster_

### Official Review · Reviewer_7d8K · 2023-10-31

**Soundness:** 3 good
**Presentation:** 2 fair
**Contribution:** 3 good
**Rating:** 5
**Confidence:** 3

**Summary:**

Most of the causal discovery approaches assume sufficiency constraint of the causal factors; this work aims to go beyond that, identifying the latent causal factors and the relation between them and observed features. Resulting in learning a full causal graph with observed and unobserved latent structures.

**Strengths:**

- The authors do a great job of setting up the motivation and providing an exhaustive list of related works
- The paper attempts to address the important problem in causal discovery

**Weaknesses:**

- Linear relations: authors make a linearity assumption between latent-latent, latent-observed, and observed-observed features. The linearity assumption between observed-observed features seems very strong
- Generalised pure pair seems to be a very strong requirement; how do you ensure this is followed in practice? For that matter, how do you test for this?
- Algorithmic description is missing; please provide a detailed description of the algorithm along with the intuition for the steps
- I understand the limited space constraint, but it would be nice to have some details on the experimental setup and the results in the main paper and discuss their implications
- The proposed method for finding all the generalised pairs is strongly dependent on relations between latent and observational factors being linear, which is not that convincing. It would be nice to see some discussion on relaxing this behaviour or ways to test this behaviour of real datasets
- It would be useful to consider synthetic datasets with known latent nodes and their interactions and compute the standard metrics in causal discovery like SHD, SID

- Presentation concerns: please introduce all the assumptions before referencing them for easier readability
- The paper is not self-contained; please consider including a background section briefly describing previous methods which are directly involved in your framework

**Questions:**

Please refer to weakness section

---

> ### Author Response · Authors · 2023-11-17
> **Response to Reviewer 7d8K (1/2)**
>
> We are very grateful for your time and effort in reviewing our paper and your constructive comments.  We hope the following new clarifications can address your concerns. Please let us know if there is any unclearness requiring further clarification.
>
> **Q1**: Concerns about the linear causal relations.
>
> **A1**:We have added some explanations about this point after the definition of the linear latent variable model (Definition 1) in Section 2.
>
> First, most existing works about latent causal structure learning focused on the linear case. Although few works can avoid this assumption, they typically have other significant limitations. For instance, [1] allows non-linearity, but it requires that all latent variables are discrete; a contemporaneous work [2] allows non-linearity, but it forbids causal edges between any two variables that share at least one common parent.
>
> Second, some theoretical results in our paper can actually generalize to two specific non-linear cases. We provided a detailed discussion in Appendix C.1. Roughly speaking, thanks to the Extended Trek Separation Theorem [3], all results in Section 3.1 and Section 3.2 are still valid as long as all causal relations involving generalized pure children (pure children and variables in a pseudo-pure pair) are still linear. Besides, if the underlying causal model follows a specific post-nonlinear model [4], we find that observed variables follow a nonparanormal distribution [5] and all theoretical results in Section 3.1 and Section 3.2 can still hold.
>
> Finally, we agree that nonlinear is common in the real world and we list the problem of causal discovery for nonlinear cases as our future work.
>
> **Q2**; Concerns about generalized pure pairs.
>
> **A2**: Considering that many previous works mentioned in our paper assume that each variable has at least two or three pure children, we politely disagree that our requirement for a generalized pure pair is very strong since two pure children are exactly a special case of a generalized pure pair. In other words, **our requirement for a generalized pure pair is much weaker compared to previous requirement for pure children**. Pure children are believed to exist because sparsity of causal edges holds in many fields such as biology and physics [6].
>
> In fact, testing assumptions about the structure of the underlying causal model is a quite difficult problem, especially those involving latent variables. For instance, even the casually sufficiency assumption that there exists no latent variable cannot be tested in most cases, let alone more complicated assumptions about pure children or generalized pure pairs. It may be possible to test this assumption with some auxiliary information (such as interventional data), but this problem is out of our current scope, we will leave it to future work.
>
> By the way, there really exists some works that can recover the latent causal structure without pure children or generalized pure pairs, but they require other potentially stronger assumptions. An example is [7], whose assumption implies that latent variables have a sufficient number of observed children, the difference between latent variables in the number of observed children is small enough, any two latent variables do not share too many common observed variables, etc.
>
> **Q3**: Concerns about experimental results.
>
> **A3**: We have moved the most representative experimental results to the main text as Section 6, which indicates that our algorithms outperform others when none of the non-Gaussianity, purity, and two-pure-children holds. Generally, SID and SHD are used in the case where there exists no latent variable [8,9]. In Section 6, since only our algorithms can recover the whole causal structure involving both latent and observed variables while others can only recover the latent causal structure, we only report the SHD of our algorithms, the average of which is 3.9 and 5.5 on $\mathcal{G}_1$ and $\mathcal{G}_2$ with sample size $N=2000$. In fact, the evaluation metrics used in our paper can sufficiently demonstrate the superiority of our algorithms. Our algorithms achieve 0% and 20% error rate on $\mathcal{G}_1$ and $\mathcal{G}_2$ while others always achieve 100% error rate. This means that in an ideal case where PC-MIMBuild makes no mistake, our algorithms can recover the correct latent causal structure in all cases for $\mathcal{G}_1$ and in 80% cases for $\mathcal{G}_2$ while other algorithms always return wrong results. In Appendix A.1, we have provided extensive experimental results on many other causal graphs. In particular, we also consider the scenario where there exists no latent variable. We are not sure what "synthetic datasets with known latent nodes" means, could you please provide more explanations? Last but not least, please note that the main contributions of our paper lie in our exhaustive theoretical analyses rather than empirical results.

---

> ### Author Response · Authors · 2023-11-17
> **Response to Reviewer 7d8K (2/2)**
>
> **Q4**: Algorithmic description is missing.
>
> **A4**: Thanks for pointing this out. We have provided detailed versions of our algorithms in Appendix D and illustrative examples to show how each step proceeds.
>
> **Q5**: Concerns about presentation.
>
> **A5**: Thanks for this constructive suggestion. We have introduced all previously used assumptions (including the three-pure-children, two-pure-children, non-Gaussianity, and purity assumption) in the second paragraph of the Introduction. Besides, our proposed assumptions are formulated as Assumption 1, 2, 3 respectively.
>
> **Q6**: The paper is not self-contained.
>
> **A6**: We have added a paragraph at the end of Section 5 (Relation to Existing Work) to briefly describe previous methods that are involved in our framework, including the Tetrad Representation Theorem [10] (It builds a connection between the structure of underlying causal structure and the covariance of variables that can be calculated from samples), Darmois-Skitovich Theorem [11] (As long as two variables share any non-Gaussian, independent component, they cannot be statistically independent), and PC-MIMBuild [12] (an overview is shown as Algorithm 4). All of them were proposed before 2010 and serve as the cornerstones of many other works. However, other more recent results they derived cannot be directly adopted in our framework where the required assumptions are not satisfied.
>
> **Reference**
>
> [1] Kivva, Bohdan, et al. "Learning latent causal graphs via mixture oracles." NeurIPS 2021.
>
> [2] Kong, Lingjing, et al. "Identification of Nonlinear Latent Hierarchical Models." NeurIPS 2023.
>
> [3] Spirtes, Peter L. "Calculation of entailed rank constraints in partially non-linear and cyclic models." UAI 2013.
>
> [4] Kaltenpoth, David, and Jilles Vreeken. "Nonlinear Causal Discovery with Latent Confounders." ICML 2023.
>
> [5] Harris, Naftali, and Mathias Drton. "PC algorithm for nonparanormal graphical models." JMLR 2013.
>
> [6] Zheng, Yujia, Ignavier Ng, and Kun Zhang. "On the identifiability of nonlinear ica: Sparsity and beyond." NeurIPS 2022.
>
> [7] Anandkumar, Animashree, et al. "Learning linear bayesian networks with latent variables." ICML 2013.
>
> [8] Brouillard, Philippe, et al. "Differentiable causal discovery from interventional data." NeruIPS 2020.
>
> [9] Mian, Osman A., Alexander Marx, and Jilles Vreeken. "Discovering fully oriented causal networks." AAAI 2021.
>
> [10] Spirtes, Peter, Clark N. Glymour, and Richard Scheines. Causation, prediction, and search. MIT press, 2000.
>
> [11] Kagan, A. M. "New classes of dependent random variables and a generalization of the Darmois–Skitovich theorem to several forms." Theory of Probability & Its Applications, 1989.
>
> [12] Silva, Ricardo, et al. "Learning the Structure of Linear Latent Variable Models." JMLR 2006.

---

> ### Author Response · Authors · 2023-11-19
> **Looking forward to your reply**
>
> Dear Reviewer 7d8K:
>
> Thanks again for your efforts in reviewing. We are looking forward to your reply. Would you mind checking our response and confirming if there are unclear explanations?
>
> Best,
> Authors

---

> ### Author Response · Authors · 2023-11-22
> **Looking forward to your reply before deadline**
>
> Dear Reviewer 7d8K:
>
> Thanks again for your time and labor in reviewing our paper. We have tried our best to address your concerns, including but not limited to more theoretical analyses (e.g., Appendix C.1), more experimental results (e.g., Section 6 in the main text, Appendix A.1), and more detailed explanations (e.g., Section 5 in the main text, Appendix D). Benefiting from your suggestions, we believe that our paper is self-contained enough. As the deadline for rolling discussion (Nov 22nd) approaches, we would be happy to take this opportunity to have more discussions. If our rebuttal has addressed your concerns, we would be grateful if you could re-evaluate our paper. Thanks!
>
> Best, Authors

---

> > ### Comment · Reviewer_7d8K · 2023-11-23
> >
> > Thank you very much for your thoughtful rebuttal.
> >
> > However, I still feel the linearity assumption between the observation-observation variable is a bit strong. Given there are many causal discovery approaches (with sufficiency assumptions) that do not make this assumption.
> >
> > As the entire proposed method builds on top of this assumption, it would be really helpful to see more discussion on non-linear relations between observed variables.
> >
> > Based on the other experimental and presentation changes, I've increased my score.

---

> ### Author Response · Authors · 2023-11-23
> **Thanks for your reply!**
>
> Dear reviewer 7d8K,
>
> Thanks very much for your kind reply. We are glad that our rebuttal has addressed most of your concerns, and we are also happy to give more discussion on nonlinear relations between observed variables.
>
> We agree that many causal discovery approaches (with sufficiency assumptions) don't rely on the linearity assumption. However, please note that recovering the whole causal structure involving both latent and observed variables is much more challenging than causal discovery without latent variables. Besides, most previous works about latent causal structure learning [2,3,4,5,6] directly employed the purity assumption that there is no causal relation between observed variables, and some studies avoiding the purity assumption [1,7] also assumed linear causal relations between observed variables.
>
> Nonetheless, we still delve deeper into non-linearity and provide more discussions in Appendix C.1. Here we present two important conclusions.
>
> 1. If only causal relations between $O_1$ and $O_2$ are nonlinear where {$O_1,O_2$} $\subset \mathbf{O}$ is not a pseudo-pure pair, then all theoretical results in Section 3.1, 3.2, and 3.3 still hold.
>
> 2. If all causal relations are nonlinear but follow a special post-nonlinear model defined as Equation (28), then all theoretical results in Section 3.1 and 3.2 still hold.
>
> We hope the above results can address your concerns, at least partially. Finally, please note that the main focus of our work still lies in the "linear latent variable models". After all, even the linear problems with latent variables have not been completely solved. Therefore, non-linearity is somewhat out of our scope, and we list it as our future work. Anyway, thanks again for your constructive suggestions and kind feedback!
>
>
> Reference
>
> [1] Silva, Ricardo, et al. "Learning the Structure of Linear Latent Variable Models." JMLR 2006.
>
> [2] Cai, Ruichu, et al. "Triad constraints for learning causal structure of latent variables." NeurIPS 2019.
>
> [3] Xie, Feng, et al. "Generalized independent noise condition for estimating latent variable causal graphs." NeurIPS 2020.
>
> [4] Xie, Feng, et al. "Identification of linear non-gaussian latent hierarchical structure." ICML 2022.
>
> [5] Huang, Biwei, et al. "Latent hierarchical causal structure discovery with rank constraints." NeurIPS 2022.
>
> [6] Chen, Zhengming, et al. "Some General Identification Results for Linear Latent Hierarchical Causal Structure." IJCAI 2023.
>
> [7] Xie, Feng, et al. "Causal discovery of 1-factor measurement models in linear latent variable models with arbitrary noise distributions." Neurocomputing 2023.

---

### Official Review · Reviewer_y7UL · 2023-10-31

**Soundness:** 3 good
**Presentation:** 3 good
**Contribution:** 3 good
**Rating:** 6
**Confidence:** 2

**Summary:**

The paper studies new identifiability criteria for linear SCMs with latent variables. They generalize several prior works on identifiability of latent variable SCMs.

**Strengths:**

- The paper seems to generalize previous results in several directions.
- As latent confounding is ubiquitous, an important problem is adressed.
- The presentation appears to be sound and clear. I did not check proofs after C.7 but until then, I found no issues.

**Weaknesses:**

- Careful proofreading would improve the paper further
- While there are several different results in the paper, it would be great if a common theme could be established, e.g., a more systematic understanding of the right identifiability conditions.
- This might be a bit much to ask, but can the conditions be tested?
- The graphical conditions are still quite restrictive (but probably unavoidable). An interesting direction might be to consider when can we answer some 'causal queries' without full identifiability under milder assumptions.


Overall, I don't have major complaints, but I am a bit hesitant because I am not too familiar with the field.

**Questions:**

- First, part of Section 3 was not very helpful for me because the results are not established at this point.

- The definition of the set $\mathbb{S}$ seems to be missing. (only in Algorithm it becomes clear what it is).

- I could not follow Assumption 2. Maybe this can be clarified. Does it mean $Nei(L)\geq 4$ and, moreover, in the case where $Nei(L)=4$ such that there is one latent and three observed neighbours then there is a pure pair....?

- Why does each variable need a latent parent? It would be nice to generalize the case without latent variables.

- 'All assumptions allow\textbf{s}'

-'It is rather milder'

- Did you normalize the data? I think it should be (I should not make a huge difference but potentially for threshold choices in your algorithm)

---

> ### Author Response · Authors · 2023-11-17
> **Response to Reviewer y7UL**
>
> We are very grateful for your time and effort in reviewing our paper and your kind encouragement.
>
> **Q1**: Careful proofreading would improve the paper further
>
> **A1**: We have tried our best to avoid typos in the proof and also invited some colleagues to check the proof. We will continually improve the proof before submitting the camera ready.
>
> **Q2**: Lack of a more systematic understanding of the right identifiability conditions.
>
> **A2**: Thanks for this constructive suggestion. We have provided some explanations in the first paragraph of Section 3.4. Specifically, we formulate two cases where none of the non-Gaussianity, purity, and two-pure-children assumption holds. The two cases both allow causal edges between observed variables but make different trade-offs between graphical and distributional assumptions. On the one hand, Case I which allows completely arbitrary distribution is more general than Case II requires partial non-Gaussianity. On the other hand, Case II which requires no pure child is more general than Case I which requires one pure child per latent variable.
>
> **Q3**: Can the conditions be tested?
>
> **A3**: We have proposed an expedient in the second paragraph of Section 3.4. Roughly speaking, Assumption 2 allows arbitrary distribution while Assumption 3 requires partial non-Gaussianity, we test whether some specific variables are all non-Gaussian. If they are, we choose Algorithm 1+3, otherwise, we choose Algorithm 1+2. Although this expedient is not a perfect method, it is better than most previous works that directly circumvented this problem. In fact, testing assumptions is a quite hard problem with only observation data, especially when there exist latent variables. On the other hand, this is really an important problem and we list it as our future work in Section 7.
>
> **Q4**: The graphical conditions are still quite restrictive (but probably unavoidable) ……
>
> **A4**: We agree that some causal questions can be answered without full identification of the causal graph. This is really an interesting research problem and we list it as our future work.
>
> **Q5**: Part of Section 3 was not very helpful for me because the results are not established at this point.
>
> **A5**: To solve this problem, we have moved some subsidiary theoretical results to Appendix.
>
> **Q6**: The definition of the set \mathbb{S} seems to be missing.
>
> **A6**: Thanks for pointing this out, we define it after Definition 5 (definition of generalized pure pair) in the revised manuscript.
>
> **Q7**: I could not follow Assumption 2. Maybe this can be clarified.
>
> **A7**: We are sorry that this causes confusion. We have clarified it in the new manuscript. Assumption 2 means that if a latent variable L has exactly one latent parent L’, three observed children $O_1$, $O_2$, $O_3$, and two of which (without loss of generality, $O_1$ and $O_2$) are its pure children, then the neighbors of $O_3$ are required not to be exactly $L$ and $L’$.
>
> **Q8**: Why does each variable need a latent parent? It would be nice to generalize the case without latent variables.
>
> **A8**: Thanks for this constructive suggestion. This inspires us to relax assumptions further, allowing some observed variables to have no latent parent. We introduce a new concept “candidate variable” as Definition 6 to prove that all our theoretical results still hold. Besides, we also provide some experimental results in Appendix A for this point, which shows that if there exists no latent variable in the underlying causal model, our algorithms return no latent variable while others may falsely introduce a latent variable.
>
> **Q9**: Did you normalize the data?
>
> **A9**: Yes, following most previous works, we normalize each variable to let them have zero mean.

---

> > ### Comment · Reviewer_y7UL · 2023-11-22
> >
> > I read the response and this clarified my questions. I appreciate the efforts to improve the clarity of the paper. However, I feel that the presentation can be improved further, e.g., it is good to illustrate why assumptions are necessary, but the new explanations are a bit difficult to follow and the edits in general are (naturally) a bit ad-hoc. Thus, I think that, given the available options, the current rating is still appropriate.
> >
> > - I meant standardize instead of normalize, i.e., unit variance of the variables to avoid varsortability bias (Reisach et al., 2021)

---

> ### Author Response · Authors · 2023-11-22
> **Further response to reviewer y7UL**
>
> We are glad that we have addressed your concerns. We have further revised the explanations about the necessity of our assumptions. In fact, to completely understand why assumptions are necessary, readers should be familiar with some concepts (such as trek and choke point) and theorems (such as the Tetrad Representation Theorem) that we defer to the Appendix due to the space limit of the main text. We only provide some high-level explanations in the main text, if you want to completely understand why assumptions are required, please refer to the proofs in the Appendix. By the way, Reviewer nXRg also agreed that "fitting everything into the page limit was a challenge". Besides, we have standardized all variables when performing experiments. Anyway, thanks for maintaining your positive score!

---

### Official Review · Reviewer_YYCe · 2023-11-01

**Soundness:** 3 good
**Presentation:** 4 excellent
**Contribution:** 3 good
**Rating:** 8
**Confidence:** 3

**Summary:**

Expanding on previous work on causal discovery for linear latent models, this work allows for causal relationships between observed children of latent variables. Specifically, they give 2 conditions under which such models are identifiable. One under which the latent has to have at least one pure child and sufficient number of neighbours. Another under which children of latents with a causal edge are allowed given that they are non-Gaussian distributed. With these weaker assumptions when compared to previous works, the authors devise an algorithm that can identify latent linear models.

**Strengths:**

- Really well written and structured
- Claims are well justified and theory is very neat.

**Weaknesses:**

- The main weakness would be the empirical evaluation. Currently, the results are only for 10 models, and the standard deviation of the accuracy is not reported. A lot of the methods discussed are not compared against. For example: Huang et al. (2022), Xie et al. (2023b) just to name a few.
- The fact that the experiments had to be deferred to the Appendix is an indication that this paper is too long already.

Minor points:
- It would be useful to readers to have graphical depictions for what causal graphs your algorithm does and does not allow. To further delineate from previous works, it may be useful to have depictions of graphs that previous works allow as well.

**Questions:**

- Given that the assumptions required are mostly on latent variables, is it possible at all to get an indication of when these assumptions might hold given a dataset?

---

> ### Author Response · Authors · 2023-11-17
> **Response to Reviewer YYCe**
>
> Thanks for your time and effort in reviewing our paper and your kind encouragement.
>
> **Q1**: The main weakness would be the empirical evaluation.
>
> **A1**: We have reported the standard derivation and added more experimental results in Appendix A. We draw 10 sample sets for each graph following previous works [1,2,3]. We haven’t included some methods in our experiments because it can be expected that these methods cannot yield correct results when none of the non-Gaussianity, purity, and two-pure-children assumption holds theoretically, and their codes have not been publicly open-sourced. Also, our main contributions are theoretical rather than empirical.
>
> **Q2**: The experiments had to be deferred to the Appendix is an indication that this paper is too long already.
>
> **A2**: we have moved some subsidiary theoretical results to Appendix and moved the most representative experimental results from Appendix to the main text.
>
> **Q3**: It would be useful to readers to have graphical depictions for what causal graphs your algorithm does and does not allow. To further delineate from previous works, it may be useful to have depictions of graphs that previous works allow as well.
>
> **A3**: Thanks for this constructive suggestion. We have already shown $\mathcal{G}_1$ and $\mathcal{G}_2$ in Figure 1 which our algorithms can handle and previous works cannot handle. Furthermore, we have also displayed $\mathcal{G}_3$~$\mathcal{G}_8$ in Figure 3. Specifically, $\mathcal{G}_3$ satisfies the three-pure-children assumption (BPC and FOFC can also yield correct results theoretically), $\mathcal{G}_4$ violates Assumption 2(a), $\mathcal{G}_5$ violates Assumption 2(b), $\mathcal{G}_6$ satisfies the non-Gaussianity, purity, two-pure-children assumptions (GIN can also yield correct results theoretically), $\mathcal{G}_7$ violates Assumption 3(a), and $\mathcal{G}_8$ violates Assumption 3(b).
>
> **Q4**: Given that the assumptions required are mostly on latent variables, is it possible at all to get an indication of when these assumptions might hold given a dataset?
>
> **A4**: It is well-known that testing assumptions with only observational data is quite difficult, especially when there exist latent variables. In Section 3.4, we only provide an expedient to guide the choice between Algorithm 2 and Algorithm 3, based on the difference between Assumption 2 and Assumption 3 in the requirement for non-Gaussianity. Although this is not a perfect method, it is better than most previous works that directly circumvented the problem of testing assumptions. This is really an important problem and we list it as our future work in Section 7.
>
>
> **Reference**
>
> [1] Silva, Ricardo, et al. "Learning the Structure of Linear Latent Variable Models." JMLR 2006.
>
> [2] Cai, Ruichu, et al. "Triad constraints for learning causal structure of latent variables." NeruIPS 2019.
>
> [3] Xie, Feng, et al. "Generalized independent noise condition for estimating latent variable causal graphs." NeurIPS 2020.

---

> > ### Comment · Reviewer_YYCe · 2023-11-22
> > **Response**
> >
> > Thanks for your response, I will keep my score the same,

---

> > > ### Author Response · Authors · 2023-11-22
> > > **Further response to Reviewer YYCe**
> > >
> > > Thanks for your reply! we are glad that you keep your already positive score.

---

### Official Review · Reviewer_nXRg · 2023-11-02

**Soundness:** 3 good
**Presentation:** 2 fair
**Contribution:** 3 good
**Rating:** 5
**Confidence:** 3

**Summary:**

The authors address the problem of causal discovery in the presence of latent variables.  As opposed to most approaches that aim to handle latent variables, which focus on estimating the causal dependence among the observed variables in the presence of latent variables, the authors focus on identifying the latent variables and the dependence structure between them and the observed variables.  Other similar methods in the literature have stricter assumptions, generally requiring either at least two pure children of each latent variables or non-Gaussianity.  The authors' proposed method has milder assumptions, requiring at a minimum only a generalized pure child pair and a latent neighbor for each latent variable.  The authors go on to describe increasing sets of assumptions that allow for stronger identifiability and present the algorithm PC-MIMBUILD.

**Strengths:**

The paper addresses an interesting problem and the initial motivation is strong.  Rather than presenting a single set of relaxing assumptions, I appreciate the authors' presentation of multiple sets that can be used depending on the situation.  The writing is generally good, and the notation is consistent and clear.  In addition, as someone largely new to this specific sub-field of research, the authors do a good job at summarizing the state of the art and positioning their work within it.

**Weaknesses:**

I have two primary concerns about this paper: the awkward presentation order of the narrative and the lack of any empirical results.

Throughout, this paper has an issue with referencing terms and concepts extensively before actually defining them.  While some level of this is often inevitable (e.g., you're not going to have defined everything by the introduction, yet you still need to give a high level overview), this is rampant enough in this paper that it often hinders understanding.  For example, in the introduction, the mentions of a "pure child" in the second paragraph, while a bit unclear, were largely fie (despite not knowing what a "pure child" was on a first read, I could at some level get that a "two-pure-children assumption" was lighter than a "three-pure-children assumption".  However, the fourth paragraph of the introduction continues to talk about "the purity assumption" and "pure children", without even trying to provide a high-level intuition of these terms.  Similarly, the fifth paragraph of the introduction has the line "the two structures in Figure 2 cannot be discriminated against each other, where {O3, O4) in Figure 2(a) is called a pseudo-pure pair." - this is essentially meaningless with any basic terminology or intuition.  I'd recommend keeping the introduction at a higher level and providing some of this discussion after Section 2.

This pattern persists, on a much smaller scale, throughout Section 3.  For example, in Section 3.1, the paragraph before Theorem 1 reads as though it assumes some familiarity with Theorem 1.  Presenting them the theorem and then presenting a description of the intuition afterwards would lead to a much clearer flow.

In general, Section 3 is high on technical detail and low on intuition.  Some of the assumptions (e.g., |Nei(L)| >= 4 in Assumption 2) feel very specific and arbitrary.  While I'm sure there's a strong motivation behind it, no discussion is provided to help the reader along in understanding where it comes from.  I'm sure that fitting everything into the page limit was a challenge, but as it stands, I think the lack of discussion and intuition makes the paper challenging to follow.

I see that empirical results are present in the appendix.  However, I wish at least some experiments were included in the main paper.  For those experiments, it also appears (though I may be mistaken) that all of the synthetic data graphs conform to the assumptions made by the authors.  I'd be interested to see how the author's approach performs under mild assumption violations, and how it compare to other algorithms when all of them have their assumptions violated in different ways.

This is a minor point and doesn't contribute to my score, but the authors use the phrase "relatively milder" a lot in this paper, and it feels awkward.  Sometimes, as in the first paragraph of Section 3, an assumption is referred to as "relatively milder" when no alternative is being discussed, leaving the comparative "milder" feeling strange.

**Questions:**

Section 2, in the second sentence, says that no observed variable can be a parent of a latent one.  This seems like a non-trivial assumption, yet it is not treated as an assumption, or justified, anywhere that I can see.  Is this a standard assumption in this sort of latent-variable identification literature?  Is there a reason to believe that, in practice, that this assumption is reasonable?

In Section 3.1, what is "latent commission"?  I don't see it anywhere in the paper apart from this mention.

Is there any guidance for how to choose which algorithm/set of assumptions to go with?  For example, in the experimental results in the Appendix, Algorithm 1+2 or Algorithm 1+3 are chosen depending on the underlying graph.  In practice, however, we won't have access to the underlying graph.

---

> ### Author Response · Authors · 2023-11-17
> **Response to Reviewer nXRg (1/2)**
>
> Thanks for your time and effort in reviewing our paper, you have provided constructive suggestions which are really helpful to improve our manuscript.
>
> **Q1**: Throughout, this paper has an issue with referencing terms and concepts extensively before actually defining them.
>
> **A1**: In the revised Introduction, we have explained all previously used assumptions when referring to them. Besides, we have also provided an intuition of the term “pure children” and avoided mentioning “pseudo-pure children” in the Introduction. Following your suggestion, we have demonstrated our motivation at a higher level in the fourth paragraph of the Introduction and provided more discussion at the end of Section 2 (Preliminaries), after defining the “pseudo-pure pair” formally. We agree that these modifications lead to easier readability.
>
> **Q2**: This pattern persists, on a much smaller scale, throughout Section 3.
>
> **A2**: Following your constructive suggestion, we have reorganized some paragraphs of Section 3. We first present our theoretical results, and then explain how they are used to identify latent variables afterwards.
>
> **Q3**: Section 3 is high on technical detail and low on intuition.
>
> **A3**: In the revised manuscript, we have provided more intuitive explanations about our assumptions. There indeed exists a strong motivation behind these assumptions, since without them, the soundness of our theoretical results is not guaranteed. For each assumption, besides clarifying “when it holds” after formulating it, we also explain “why we need it" at a high level after the theoretical results whose soundness heavily relies on it. For instance, we explain why we require Assumption 2(b), i.e., |Nei(L)| >= 4 after Lemma 2 and Corollary 1. We suggest that without Assumption 2(b), Lemma 2 may not hold, and take Figure 2 as an example to show this. We only provide some rough explanations in the main text. To completely understand why assumptions are necessary, readers should be familiar with some concepts (such as trek and choke point) and theorems (such as the Tetrad Representation Theorem) that we defer to Appendix due to the space limit of the main text. Therefore, if you want to get more insights, please read our proofs rather than only the main text.
>
> **Q4**: Lack of experimental results.
>
> **A4**: First, we have moved the most representative experimental results to the main text as Section 6, which indicates that our algorithms outperform others when none of the non-Gaussianity, purity, and two-pure-children holds. Moreover, following your suggestion, we have performed comparisons in many other cases including but not limited to $\mathcal{G}_3$: the three-pure-children assumption holds (BPC and FOFC can also yield correct results theoretically); $\mathcal{G}_4$: Assumption 2(a) is violated; $\mathcal{G}_5$: Assumption 2(b) is violated; $\mathcal{G}_6$: the non-Gaussianity, purity, two-pure-children assumptions all hold (GIN can also yield correct results theoretically); $\mathcal{G}_7$: Assumption 3(a) is violated; $\mathcal{G}_8$: Assumption 3(b) is violated. BPC, FOFC, and our algorithms can all achieve low error rates on $\mathcal{G}_3$, GIN and our algorithms can achieve low error rates on $\mathcal{G}_6$ and ours is more robust than GIN, all algorithms cannot yield correct results on other graphs. The detailed experimental results are provided in Appendix A, and we also analyze why our algorithms yield such results.
>
> **Q5**: The usage of “relatively milder”.
>
> **A5**: We use this phrase because Assumption 2 is strictly milder than the three-pure-children assumption required, and Assumption 3 is strictly milder than a combination of the non-Gaussianity and two-pure-children assumption. Following you suggestion, we have reduced the frequency of its usage.

---

> ### Author Response · Authors · 2023-11-17
> **Response to Reviewer nXRg (2/2)**
>
> **Q6**: Concerns about the assumption that no observed variable can be a parent of a latent one.
>
> **A6**: Since proposed by the seminal work, this assumption (called measurement assumption by some researchers) has almost become a standard assumption for latent causal structure learning. All related works mentioned in our paper made this assumption except [1], which has other significant limitations in that their algorithms are only theoretically sound but computationally intractable. This assumption is believed to hold because generally, observed variables are low-level, concrete objects while latent variables are high-level, abstract concepts [2]. In most cases, the former influence the latter but not vice versa. On the other hand, it is also possible that this assumption is not satisfied, so in Section 7, we have listed it as our future work.
>
> **Q7**: What does “latent commission” mean?
>
> **A7**: We are sorry that this term causes confusion. We have removed it from the main text except for Section 6 (Experimental Results), where “latent commission” serves as an evaluation metric that is the redundant latent variables divided by the total number of latent variables in the ground truth graph.
>
> **Q8**: Is there any guidance for how to choose which algorithm/set of assumptions to go with?
>
> **A8**: We have proposed an expedient for choosing different algorithms in Section 3.4 of the revised manuscript. Roughly speaking, Assumption 2 allows arbitrary distribution while Assumption 3 requires partial non-Gaussianity, we test whether some specific variables are all non-Gaussian. If they are, we choose Algorithm 1+3, otherwise, we choose Algorithm 1+2. Although this expedient is not a perfect method, it is better than most previous works that directly circumvented this problem. In fact, testing assumptions is a quite hard problem with only observational data, especially when there exist latent variables. On the other hand, this is really an important problem and we also list it as our future work in Section 7.
>
>
> **Reference**
>
> [1] Adams, Jeffrey, Niels Hansen, and Kun Zhang. "Identification of partially observed linear causal models: Graphical conditions for the non-gaussian and heterogeneous cases." NeurIPS 2021.
>
> [2] Schölkopf, Bernhard, et al. "Toward causal representation learning." Proceedings of the IEEE 2021.

---

> ### Author Response · Authors · 2023-11-19
> **Looking forward to your reply**
>
> Dear Reviewer nXRg:
>
> Thanks a lot for your time and efforts in reviewing this paper. We have tried our best to address the mentioned concerns. Are there unclear explanations and descriptions here? We could further clarify them.
>
> Best,
> Authors

---

> ### Author Response · Authors · 2023-11-22
> **Looking forward to your reply before deadline**
>
> Dear Reviewer nXRg:
>
> Thanks again for your constructive suggestions. We have tried our best to address your concerns. For instance, we have reorganized our paper to improve the presentation and also provided more extensive experimental results in not only the main text but also the appendix. We believe that these modifications improve our manuscript significantly. As the deadline for rolling discussion (Nov 22nd) approaches, we would be happy to take this opportunity to have more discussions. If our rebuttal has addressed your concerns, we would be grateful if you could re-evaluate our paper. Thanks!
>
> Best, Authors

---

### Author Response · Authors · 2023-11-17
**General Response to Reviewers**

We sincerely thank all reviewers for their insightful comments. Inspired by their advice, we have added more theoretical and experimental results and also tried our best to improve the presentation. The main changes in our manuscript are summarized as follows.

1. We have added more intuitive explanations to make our theoretical results easier to understand. For instance, besides clarifying when our required assumptions hold (after Assumption 1, Assumption 2, and Assumption 3), we also explain why these assumptions are required (after Theorem 1, Corollary 1, and Corollary 2).

2. We have further relaxed our required assumptions. For instance, we prove that some of our theoretical results can generalize to nonlinear cases (details are provided in Appendix C.1), and we allow that some observed variables have no latent parent (for which we have introduced a new concept “candidate variable” in Definition 6). Also, Assumption 3 has been adjusted very slightly (In this way, Assumption 3 is strictly weaker than a combination of the non-Gaussianity assumption and the two-pure-children assumption).

3. We have provided more experimental results. For instance, we display the performance of our proposed algorithms when some required assumption is not satisfied and also analyze why our algorithms return such results. Details are provided in Appendix A.

4. We have tried our best to improve the presentation. For instance, we have moved some subsidiary theoretical results to Appendix and moved the most representative experimental results from Appendix to the main text. Besides, we have adjusted the organization of some paragraphs to make sure that most concepts and theorems are introduced before being referred to for easier readability.

We marked major changes in the revised manuscript in a violet font.

---

### Meta-Review · Area_Chair_ZA6q · 2023-12-08

**Metareview:**

The contribution advances the literature on learning (linear) causal structure from data. Its core idea is to explore a different combination of assumptions regarding distributional (some non-Gaussianity, but less than in other results) and structural (some type of sparsity, but less than in other results) aspects. It is carefully done and deserves a readership - I have read the paper, particularly given the relatively lack of reply to authors - but it is also true that the contributions are on the subtle side and likely to better appreciated by expert readers.

**Justification For Why Not Higher Score:**

It is relatively narrow in scope, given that it fills up a specific gap regarding the combinations of two kinds of assumptions (distributional and structural) within a body of research which is already relatively mature.

**Justification For Why Not Lower Score:**

It is correct to say that people in this area would be interested in the results of this contribution, which is competently carried out.

---

### Decision · Program_Chairs · 2024-01-16

Accept (poster)